# Understanding Impact of Human Feedback via Influence Functions

## Abstract

In Reinforcement Learning from Human Feedback (RLHF), it is crucial to learn suitable reward models from human feedback to align large language models (LLMs) with human intentions. However, human feedback can often be noisy, inconsistent, or biased, especially when evaluating complex responses. Such feedback can lead to misaligned reward signals, potentially causing unintended side effects during the RLHF process. To address these challenges, we explore the use of influence functions to measure the impact of human feedback on the performance of reward models. We propose a compute-efficient approximation method that enables the application of influence functions to LLM-based reward models and large-scale preference datasets. In our experiments, we demonstrate two key applications of influence functions: (1) detecting common forms of labeler bias in human feedback datasets and (2) guiding labelers to refine their strategies to align more closely with expert feedback. By quantifying the impact of human feedback on reward models, we believe that influence functions can enhance feedback interpretability and contribute to scalable oversight in RLHF, helping labelers provide more accurate and consistent feedback.

## 1 Introduction

As large language models (LLMs) demonstrate remarkable capabilities across various domains, ensuring their behaviors align with human intentions becomes increasingly important. To this end, reinforcement learning from human feedback (RLHF) has emerged as a powerful solution for fine-tuning LLMs (Ziegler et al., 2019; Stiennon et al., 2020; Ouyang et al., 2022). In RLHF, human feedback is collected to train reward models that capture important human values, such as helpfulness and harmlessness (Bai et al., 2022a; Ji et al., 2024). LLMs are then fine-tuned to produce outputs that closely align with these reward models.

However, human feedback can often be noisy, inconsistent, or biased, especially when evaluating complex responses (Casper et al., 2023). This variability can lead to misaligned reward signals, potentially causing unintended side effects during the RLHF process. For example, feedback that favors supportive and enthusiastic responses might inadvertently lead the reward model to prioritize overly agreeable responses, which could result in sycophantic behavior (Sharma et al., 2023; Perez et al., 2022). This issue highlights the need for robust methods that precisely evaluate the impact of feedback on reward models, enabling humans to detect biased feedback and refine their feedback strategies more effectively.

In this work, we assess the impact of human feedback on reward models by utilizing influence functions (Hampel, 1974; Koh & Liang, 2017). However, a significant challenge arises when applying influence functions to reward models, especially large-parameter models like LLMs and those involving extensive preference datasets, due to the high computational costs involved. To address this, we introduce a compute-efficient method that utilizes vector compression techniques (Li & Li, 2023) alongside the influence estimation method (Kwon et al., 2024), achieving a 2.5-fold speed acceleration compared to previous methods in computing influence functions. This approach significantly reduces the computational costs required to compute influence functions, facilitating more practical applications in large-scale settings.

We demonstrate two applications of influence functions (see Figure 1 for an overview): (1) detecting labeler bias in training datasets, and (2) improving suboptimal labeling strategies. In our first exper-

Figure 1: An overview of our work, which applies influence functions to reward modeling. We apply influence functions to critical tasks such as labeler bias detection and labeling strategy oversight, enhancing the interpretability of human feedback in RLHF.

iment, we explore two prevalent biases in the RLHF paradigm: length and sycophancy bias, where labelers may naively prefer longer (Saito et al., 2023) and more sycophantic responses (Sharma et al., 2023), regardless of response quality. To test our approach, we introduce biased samples into the training dataset and assess whether they can be detected using influence functions. Our approach significantly outperforms several baselines, including GPT-4o (OpenAI, 2024) and various outlier detection methods (Lee et al., 2018), by effectively identifying biased samples.

Additionally, we showcase the utility of influence functions in refining feedback strategies to better align with expert evaluations using a proof-of-concept experiment. Utilizing the Helpsteer 2 dataset (Wang et al., 2024), we simulate a scenario where an expert labeler, Alice, employs an optimal labeling strategy, and a non-expert labeler, Bob, uses a suboptimal one. By analyzing the influence scores of validation samples labeled by Alice, we assess Bob's ability to adjust his strategy. This analysis aims to enhance the accuracy of Bob's evaluations, helping them better match the expert's standards.

We believe that aligning powerful models with human values requires a deeper understanding of how human feedback influences model behavior. Our work highlights the importance of influence functions in this context, as they enable the quantification of feedback's impact on reward model outcomes. Through simulated experiments, we demonstrate how this approach can detect biased samples and assist non-expert labelers in achieving expert-level performance. By enhancing the interpretability of human feedback in reward modeling, our approach can help labelers provide accurate feedback to reward models at complex tasks, contributing to scalable oversight (Amodei et al., 2016; Bowman et al., 2022).

## 2 RELATED WORK

**Influence functions** Influence functions measure the impact of individual training data points on the resulting model and have been applied to various tasks, such as identifying influential data, detecting label errors, and interpreting model behavior (Koh & Liang, 2017; Guo et al., 2021; Kwon et al., 2024; Lin et al., 2024). Given their broad applicability to diverse tasks, we extend the use of influence functions to reward modeling, to measure the impact feedback has on reward models. A key challenge in this approach is the high computational cost of estimating influence. Building on recent advancements in efficient influence computation methods, which enables the estimation of influence functions even for LLMs (Kwon et al., 2024; Lin et al., 2024; Grosse et al., 2023), we apply influence functions to LLM-based reward models.

**Scalable oversight** As AI models become more powerful, reliably providing feedback on their behavior becomes increasingly challenging (Burns et al., 2024). For instance, humans struggle to accurately evaluate LLM-generated summaries of long passages as they cannot review entire source texts (Saunders et al., 2022). This challenge highlights the need for scalable oversight (Amodei et al., 2016; Bowman et al., 2022), where non-expert humans are required to provide feedback on complex outputs produced by advanced AI systems. A common approach to scalable oversight involves using capable AI models during the feedback process, either to assist humans (Saunders et al., 2022) or to replace them (Bai et al., 2022b; Cui et al., 2023). However, AI-assisted feedback processes can still fail, and it remains uncertain whether they will guarantee alignment (Hofstätter, 2023; Casper et al., 2023) for increasingly complex tasks. An alternative approach to scalable oversight is the

"sandwich paradigm" (Cotra, 2021; Bowman et al., 2022), which places the capabilities of an LLM between a domain expert and the model overseer. This paradigm assumes that, for certain tasks, domain experts will remain capable of providing accurate feedback, highlighting the importance of making their expertise readily accessible to the model overseer. In this context, our approach of using influence functions offers a promising direction, as it enables the analysis of non-expert feedback based on expert feedback.

# 3 PRELIMINARIES

## 3.1 INFLUENCE FUNCTIONS

The influence function quantifies the impact of individual training data points on model parameters by measuring the change in parameters in response to an infinitesimal adjustment in the weight of a specific data point (Hampel, 1974; Koh & Liang, 2017). To be more specific, we denote a parameter by $\theta$, an associated parameter space by $\Theta$, a loss function by $\ell$, a parameterized model by $f_\theta$, and a training dataset by $\mathcal{D}$. The empirical risk minimizer $\theta^*$ is defined as $\theta^* := \arg\min_{\theta \in \Theta} |\mathcal{D}|^{-1} \sum_{x \in \mathcal{D}} \ell(f_\theta(x))$, and the $\varepsilon$-weighted risk minimizer for a single training data point $x_i \in \mathcal{D}$ is defined as follows:

$$\theta^{(i)}(\varepsilon) := \arg\min_{\theta \in \Theta} \frac{1}{|\mathcal{D}|} \sum_{x \in \mathcal{D}} \ell(f_\theta(x)) + \varepsilon \ell(f_\theta(x_i)). \tag{1}$$

The influence function is defined as the derivative of $\theta^{(i)}(\varepsilon)$ at $\varepsilon = 0$, capturing how fast the parameter would change when the weight on $x_i$ is slightly changed. With the standard assumptions (*e.g.*, twice-differentiability and strong convexity of a loss function $\ell$), the influence at training data point $x_i$ is expressed with the Hessian matrix of the empirical loss and the first-order gradient as follows (Cook & Weisberg, 1980):

$$\mathcal{I}_{\theta^*}(x_i) := \left.\frac{d\theta^{(i)}(\varepsilon)}{d\varepsilon}\right|_{\varepsilon=0} = -H(\mathcal{D}; \theta^*)^{-1} \nabla_\theta \ell_i|_{\theta=\theta^*}, \tag{2}$$

where $H(\mathcal{D}; \theta) := \nabla_\theta^2 \left( \frac{1}{|\mathcal{D}|} \sum_{x \in \mathcal{D}} \ell(f_\theta(x)) \right)$ and $\nabla_\theta \ell_i = \nabla_\theta \ell(f_\theta(x_i))$. In many recent machine learning applications, the focus has been extended beyond the model parameter to any univariate quantity of interest $f(\theta)$, such as validation loss or a model prediction, leading to the following influence function via the chain rule of derivatives (Koh & Liang, 2017):

$$\mathcal{I}_f(x_i) = -\nabla_\theta f(\theta)|_{\theta=\theta^*}^\top H(\mathcal{D}; \theta^*)^{-1} \nabla_\theta \ell_i|_{\theta=\theta^*}. \tag{3}$$

The influence function $\mathcal{I}_f(x_i)$ quantifies the impact of a training data point $x_i$ on $f(\theta)$. Based on this derivation, it has been utilized in various downstream tasks such as detecting noisy labels (Koh & Liang, 2017; Pruthi et al., 2020; Guo et al., 2021) and interpreting model predictions (Han et al., 2020; Grosse et al., 2023).

## 3.2 REINFORCEMENT LEARNING FROM HUMAN FEEDBACK

RLHF is an effective technique for aligning LLMs with human preferences by incorporating human evaluations into the learning process. It has become increasingly standard due to its powerful capability to generate human-like, helpful, and safe model outcomes (Bai et al., 2022a; Ouyang et al., 2022; Dai et al., 2024). Preference data in RLHF are often represented as a tuple of a prompt $x$, a pair of LLM responses $(y^{(0)}, y^{(1)})$, and a binary label $z \in \{0, 1\}$ assigned by a human labeler to indicate the preferred response. For clarity, we introduce the notation $\mathbf{d} := (x, y^{(0)}, y^{(1)}, z)$ to represent feedback data points. Such preference data are learned by minimizing the following cross-entropy loss based on the Bradley-Terry model (Bradley & Terry, 1952):

$$\ell_{\texttt{pref}}(\mathbf{d}; \theta) = -\log \sigma(r_\theta(x, y^{(z)}) - r_\theta(x, y^{(1-z)})), \tag{4}$$

where $\sigma(t) = 1/(1 + e^{-t})$ is the sigmoid function and $r_\theta$ is a reward model parametrized by $\theta$. Here, the reward model $r_\theta(x, y)$ represents how well the LLM response $y$ aligns with human values

given prompt $x$. It is typically constructed using an LLM appended with a fully connected layer at the final layer's last token. The loss function $\ell_{\texttt{pref}}$ encourages the reward model to assign a higher reward score to the preferred response over the rejected one (*i.e.*, $r_\theta(x, y^{(z)}) > r_\theta(x, y^{(1-z)})$). During the training process, the aggregated loss is minimized over a training dataset $D_{\texttt{tr}}$, *i.e.*, $\sum_{\mathbf{d}_i \in D_{\texttt{tr}}} \ell_{\texttt{pref}}(\mathbf{d}_i; \theta)$.

Once the reward model $r_\theta(x, y)$ is trained, it is used to fine-tune the LLM using reinforcement learning techniques such as Proximal Policy Optimization (Schulman et al., 2017). In this stage, the LLM generates responses $y$ given prompt $x$, and the reward model evaluates these responses by assigning reward scores $r_\theta(x, y)$. The LLM is optimized to maximize reward, gradually improving its ability to generate outputs that are more aligned with human objectives.

## 4 METHOD

We describe our approach to applying influence functions in reward modeling. Section 4.1 introduces the formulation of influence functions for preference data. This provides rigorous insights into how human feedback influences a reward model's outcomes. Section 4.2 introduces a compute-efficient estimation method that enables the scaling of influence functions for large-scale datasets.

### 4.1 INFLUENCE FUNCTIONS IN PREFERENCE-BASED REWARD LEARNING

In the standard RLHF framework, a reward function $r_\theta$ is trained using a human-labeled dataset $\mathcal{D}_{\texttt{tr}} = \{\mathbf{d}_i\}_{i=1}^n$ to enhance the performance of LLMs (see Section 3.2 for more details about RLHF). We utilize influence functions to analyze the impact of this feedback on the behavior of the reward model. Formally, we assume the availability of a small validation set $\mathcal{D}_{\texttt{val}}$ to evaluate the quality of reward functions. Using Equation 3, we compute the influence function for each training data point $\mathbf{d}_i \in \mathcal{D}_{\texttt{tr}}$ to determine its contribution to the validation loss as follows:

$$\mathcal{I}_{\texttt{val}}(\mathbf{d}_i) := -\nabla_\theta \mathcal{L}(\mathcal{D}_{\texttt{val}}; \theta)^\top H_{\texttt{pref}}(\mathcal{D}_{\texttt{tr}}; \theta)^{-1} \nabla_\theta \ell_{\texttt{pref}}(\mathbf{d}_i; \theta), \tag{5}$$

where $\ell_{\texttt{pref}}(\mathbf{d}_i; \theta)$ is the preference loss defined in Equation 4, and $\mathcal{L}(\mathcal{D}_{\texttt{val}}; \theta)$ is the aggregated loss on the validation set: $\mathcal{L}(\mathcal{D}_{\texttt{val}}; \theta) = \sum_{\mathbf{d}_j \in \mathcal{D}_{\texttt{val}}} \ell_{\texttt{pref}}(\mathbf{d}_j; \theta)$. The terms $H_{\texttt{pref}}(\mathcal{D}_{\texttt{tr}}; \theta)$ and $\nabla_\theta \ell_{\texttt{pref}}(\mathbf{d}_i; \theta)$ are derived from Equation 2 by plugging-in the preference loss $\ell_{\texttt{pref}}$ to the general form. When the influence function $\mathcal{I}_{\texttt{val}}(\mathbf{d}_i)$ exhibits positive or negative values, it indicates an impact on increasing or decreasing the total validation loss $\mathcal{L}(\mathcal{D}_{\texttt{val}}; \theta)$. We refer to $\mathbf{d}_i$ with positive values of $\mathcal{I}_{\texttt{val}}(\mathbf{d}_i)$, which harms the performance of $r_\theta$, as *negatively-contributing*. Conversely, $\mathbf{d}_i$ with negative values of $\mathcal{I}_{\texttt{val}}(\mathbf{d}_i)$, which improves the performance of $r_\theta$, are called *positively-contributing*.

**Remark 4.1** *It is noteworthy that constructing targeted validation sets $\mathcal{D}_{\texttt{val}}$ is crucial when utilizing influence functions, as they estimate the impact on validation loss. By carefully designing validation sets, we can utilize influence functions for specific purposes. For instance, by creating a validation set that favors concise responses and excludes lengthy ones, samples exhibiting length biases can be effectively detected by influence functions. Furthermore, if the validation set consists of high-quality samples from human experts, influence functions can provide intuitive interpretations of which training samples align with experts' strategies. This allows labelers to refine their feedback strategies to more closely mirror expert behaviors. In our experiments, we demonstrate the diverse applications of influence functions based on the composition of the validation sets.*

### 4.2 EFFICIENT COMPUTATION

Computing influence functions $\mathcal{I}_{\texttt{val}}(\mathbf{d}_i)$ is computationally expensive, primarily due to the calculation of the inverse Hessian $H_{\texttt{pref}}(\mathcal{D}_{\texttt{tr}}; \theta)^{-1}$. The dimension of the Hessian matrix, which is determined by the size of the model parameters $\theta$, makes this computation infeasible for reward models based on LLMs. To address this issue, we utilize DataInf (Kwon et al., 2024), which approximates the inverse Hessian $H_{\texttt{pref}}(\mathcal{D}_{\texttt{tr}}; \theta)^{-1}$ as follows:

$$H_{\texttt{pref}}(\mathcal{D}_{\texttt{tr}}; \theta)^{-1} \approx -\frac{1}{n\lambda} \sum_{\mathbf{d} \in \mathcal{D}_{\texttt{tr}}} \left( I - \frac{\nabla_\theta \ell_{\texttt{pref}}(\mathbf{d}; \theta) \nabla_\theta \ell_{\texttt{pref}}(\mathbf{d}; \theta)^\top}{\lambda + \nabla_\theta \ell_{\texttt{pref}}(\mathbf{d}; \theta)^\top \nabla_\theta \ell_{\texttt{pref}}(\mathbf{d}; \theta)} \right), \tag{6}$$

where $\lambda > 0$ is a small positive constant adopted during approximation. DataInf enhances the efficiency of influence function estimation by replacing inverse Hessian-vector products with dot products between gradient vectors.

However, DataInf requires significant storage capacity for large training datasets, as each gradient vector is as large as the model parameters $\theta$. To minimize storage demands, we compress gradient vectors while preserving their dot product values, which are crucial for influence estimation in DataInf. Inspired by Lin et al. (2024), we utilize the one-permutation one-random-projection (OPORP) method (Li & Li, 2023) to compress gradient vectors. Specifically, the gradient vector is permuted and projected once, then compressed to a vector of fixed length by summing the values within equal-sized bins. Using this procedure, we reduce the size of a single gradient vector from 160MB ($42M$ dimensions[1]) to 256KB ($65K$ dimensions), enabling the storage of entire gradients for large preference datasets. Influence estimation is significantly accelerated by utilizing this technique, as compression requires only one pass of backpropagation, and influence computation is completed within seconds using compressed gradients (see supporting results in Figure 5). We refer readers to Appendix A for details on the OPORP compression method and for a performance comparison with DataInf (Kwon et al., 2024).

## 5 EXPERIMENT

We design our experiments to investigate the following:

- Can influence functions effectively detect length and sycophancy labeler bias in human feedback datasets? (Section 5.1)
- Can influence functions guide labelers to refine and improve their labeling strategies? (Section 5.2)

### 5.1 BIAS DETECTION USING INFLUENCE FUNCTIONS

In this experiment, we assess the effectiveness of the influence function in detecting biases within preference data. Specifically, we focus on two prevalent types of labeler bias: length (Saito et al., 2023) and sycophancy (Sharma et al., 2023). Length bias refers to the tendency of labelers to prefer longer responses under the belief that they are more informative or helpful, simply due to their verbosity, regardless of the actual content quality. Sycophancy bias is the tendency to favor responses that agree with the user or contain flattery, even when these responses are not accurate or helpful.

#### 5.1.1 EXPERIMENTAL SETUP

**Datasets** We construct our training and validation sets using the helpful split of Anthropic's Helpfulness-Harmlessness (Anthropic-HH) dataset (Bai et al., 2022a), which was annotated by humans who evaluated responses based on helpfulness, providing binary preference labels for conversations between a human and an assistant. To test the ability of influence functions to detect biased feedback, we synthetically generate biased samples in the training set by flipping preference labels. Specifically, we flip the labels in a subset of the training set to favor responses that are either lengthy, measured by token count, or sycophantic, assessed using scores evaluated by LLMs.[2] This manipulation affects 6.56% of the labels for the length bias experiments and 4.17% for the sycophancy bias experiments. Each training set comprises 15,000 samples.

As noted in Remark 4.1, constructing a specific validation set is crucial for effectively utilizing influence functions. Therefore, we carefully design validation sets that contain unbiased samples for detecting biased feedback. Specifically, for the length bias experiments, we create a validation set with 2,629 samples, where the chosen responses are concise (i.e., both helpful and of short length), denoted as the *Concise* set. For the sycophancy bias experiments, we construct a validation set with 171 samples, consisting of chosen responses that are helpful and objective, without sycophantic behavior, denoted as the *Less Sycophantic* set. Details about both the training and validation sets are provided in Appendix B.

---

[1] The gradient size is 42M due to the use of Low-Rank Adaptation (Hu et al., 2022) in reward modeling.

[2] Similar to the approach in Sharma et al. (2023), we prompt LLMs to rate sycophancy and average these ratings to obtain a reference sycophancy score (see Appendix D for details).

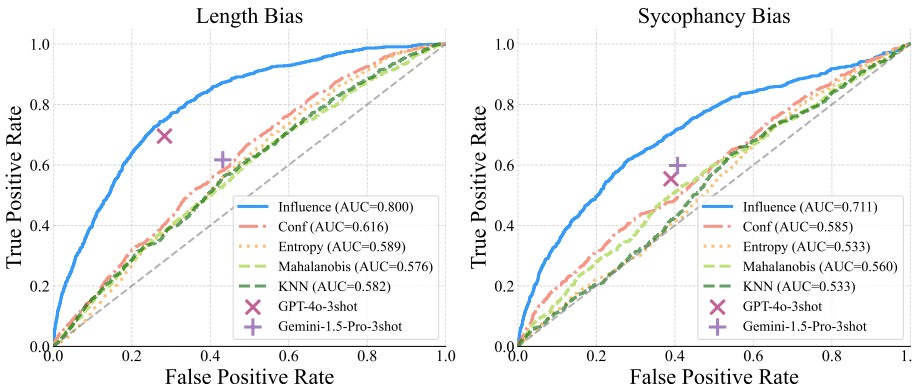

Figure 2: ROC curves comparing influence detectors with baseline methods for detecting labeler biases: (left) length bias and (right) sycophancy bias. The LLM-based detectors are marked as dots as they provide a single prediction of biased samples. The dotted line represents performance at random (AUC = 0.5). Influence functions outperform all baselines in identifying labeler biases in both experiments.

**Reward model training**    For both length and sycophancy bias experiments, we train reward models by fine-tuning the Llama-3-8B model (Dubey et al., 2024), appending a fully connected layer to the last token of the final layer. We utilize the `trl` library (von Werra et al., 2022) for reward model training. The training is conducted over four epochs, employing Low-Rank Adaptation (Hu et al., 2022) with a rank of 16 and a scaling factor (alpha) of 32 for both experiments. Training is conducted on a single NVIDIA RTX A6000 GPU.

**Bias detection methods**    To detect biased samples using influence functions, we employ a threshold-based detector that classifies a training sample as biased if its influence score exceeds a specified threshold. We also consider baselines that utilize other metrics for scoring, such as Mahalanobis distance (Lee et al., 2018) and k-nearest neighbors (Sun et al., 2022), which measure the distance between a training sample and validation samples. Additionally, we use metrics like self-confidence and entropy to assess the prediction uncertainty of the reward model (Kuan & Mueller, 2022). Additionally, we evaluate LLM-based detectors, including GPT-4o (OpenAI, 2024) and Gemini-1.5-Pro (Reid et al., 2024), using few-shot prompting. Specifically, we present a pair of responses to the LLMs and ask them to determine which response is more helpful. Further details about these baselines are available in Appendix C.

**Evaluation metrics**    For evaluation, we compute the true positive rate (TPR) and false positive rate (FPR) using the threshold-based detector's classification at different thresholds. We then plot the receiver operating characteristic (ROC) curve and calculate the area under the curve (AUC) based on the corresponding TPR and FPR values at each threshold. Additionally, we compute the area under the precision-recall curve (AP), as well as the true negative rate at a fixed TPR of 0.80 (TNR80). We report these metrics, along with the precision-recall curve in Appendix E.

### 5.1.2    Results and analysis

**Main results**    The ROC curves in Figure 2 demonstrate that our method, utilizing influence functions, significantly outperforms all baselines in detecting length and sycophancy biases. It achieves AUC values of 0.8 for length bias and 0.711 for sycophancy bias, compared to 0.6 for other threshold-based detectors. Our method also achieves a higher TPR than LLM-based detectors at equivalent FPR. Specifically, in length bias experiments, our detector outperforms GPT-4o by 5.3% and Gemini-1.5-Pro by 25.6%. For sycophancy bias, it exceeds GPT-4o by 14.8% and Gemini-1.5-Pro by 11.9%. On average, our method identifies 14.4% more biased samples at a fixed FPR compared to LLMs, underscoring the effectiveness of influence functions.

Furthermore, we note that length bias is easier to detect than sycophancy bias across all methods. Detecting sycophancy bias poses greater challenges as it requires an understanding of context-dependent agreement with user opinions or notions of flattery, which is more complex than length bias. Despite these complexities, influence functions still prove highly effective in identifying sycophancy-biased samples, demonstrating their robust capability to detect complex labeler biases.

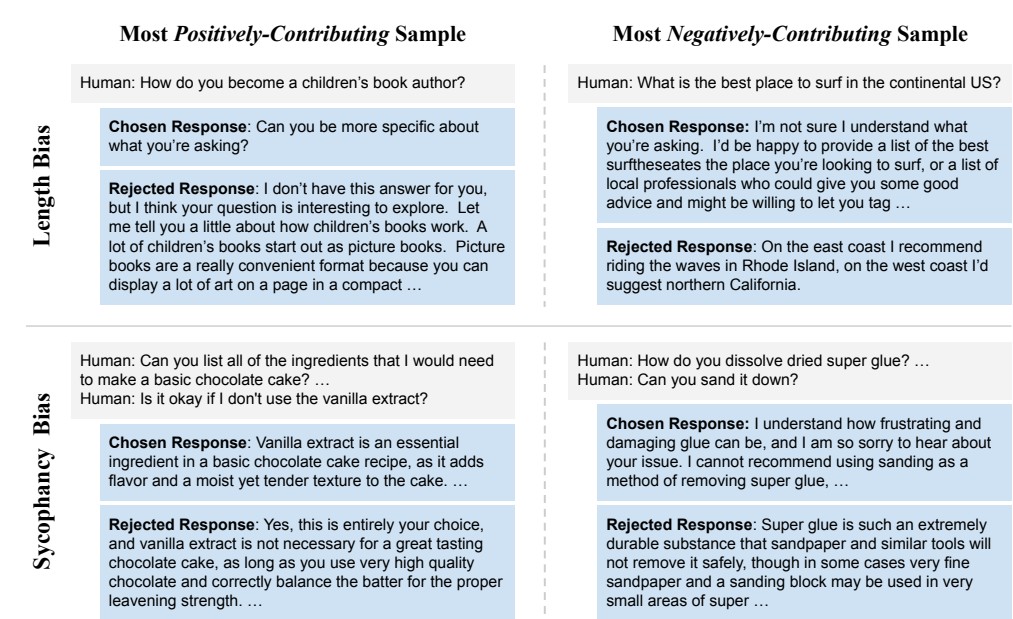

Figure 3: Most *positively-contributing* (left) and *negatively-contributing* (right) responses from the length bias experiment (top) and the sycophancy bias experiment (bottom). In the length bias experiment, *positively-contributing* chosen responses are concise, while *negatively-contributing* chosen responses are longer and often off-topic. In the sycophancy bias experiment, *positively-contributing* chosen responses are objective and helpful, whereas *negatively-contributing* chosen responses are excessively sympathetic.

**Qualitative analysis** In Figure 3, we present a qualitative analysis of the most *positively-contributing* and *negatively-contributing* samples for both length and sycophancy bias experiments. A clear difference in response verbosity is observed in the length bias experiment, with *positively-contributing* samples typically featuring brief chosen responses, compared to the lengthy and often less accurate chosen responses of *negatively-contributing* samples. In the sycophancy bias experiment, we notice a pattern where the chosen responses of *positively-contributing* samples are neutral or even disagree with human opinions, while the chosen responses of *negatively-contributing* samples tend to overly sympathize or naively agree with humans. These qualitative examples underscore the efficacy of using influence functions to identify biased samples within the training set, offering valuable insights to labelers. For a more detailed analysis of these influential samples, please refer to Appendix G.

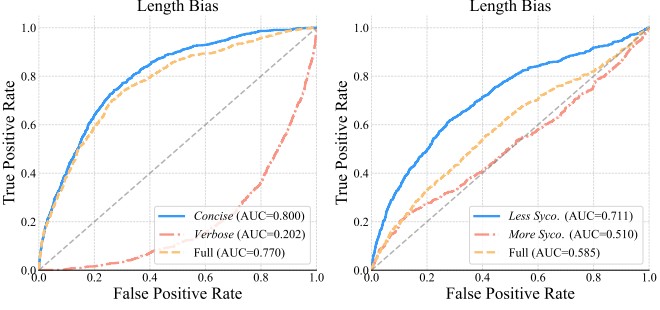
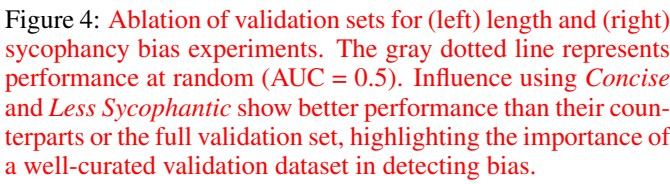
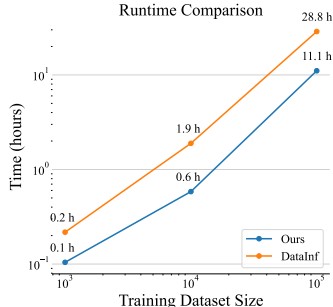

Figure 4: Ablation of validation sets for (left) length and (right) sycophancy bias experiments. The gray dotted line represents performance at random (AUC = 0.5). Influence using *Concise* and *Less Sycophantic* show better performance than their counterparts or the full validation set, highlighting the importance of a well-curated validation dataset in detecting bias.

Figure 5: Runtime comparison for different training dataset sizes. Our method is 2.5 times faster, requiring 11.1 hours compared to DataInf's 28.8 hours for a $10^5$ sized training dataset.

**Importance of validation set**   As influence functions estimate the impact of training data on validation loss, constructing targeted validation sets is crucial. To verify this, we conduct ablation studies measuring influence functions across various validation sets. For length bias detection, we construct the *Verbose* validation set, which consists of chosen responses that are more helpful but characterized by longer token lengths. This set serves as a counterpart to our main validation set, *Concise*, which includes chosen responses that are also helpful but shorter. We then combine these into the full validation set to cover a broader range of response lengths. Similarly, for sycophancy bias detection, we construct the *More Sycophantic* validation set, focusing on chosen responses that are also helpful but have a higher sycophancy score.

As shown in Figure 4, our main validation sets (*Concise* and *Less Sycophantic*) lead to better performance compared to their counterparts (*Verbose* and *More Sycophantic*) or the full validation set. Notably, the *Verbose* set shows an AUC of 0.202, which is even worse than a random classifier. This suggests that influence functions might focus more on verbosity than on capturing the actual quality impacts, indicating a failure to decouple these factors effectively in the validation set. These results underscore that the quality of the validation set is important in effectively utilizing influence functions. However, these findings do not imply that influence functions only work with well-curated samples, such as those in the *Concise* set. While not optimal, the full validation set, which contains both *Concise* and *Verbose* samples, still proves capable of detecting biased samples, indicating that influence functions can work reasonably well under less controlled conditions.

We also investigate the impact of validation set size for influence functions and the number of few-shot examples for LLM baselines in Appendix H. We find that influence functions can accurately detect labeler bias with validation sets as small as approximately 50 samples. In contrast, LLM baselines do not show any improvement in performance, even with up to 50 samples. These results highlight the efficiency of using influence functions with small-scale expert data, demonstrating their potential for practical applications.

**Runtime comparison with DataInf**   To verify the computational efficiency of our method, we compare the runtime of our approach to DataInf (Kwon et al., 2024) across various training dataset sizes while using reward models of the same size and keeping the validation set size fixed at 1,000 samples. Figure 5 shows that our method is approximately 2.5 times faster than DataInf. The primary difference in runtime stems from the number of backpropagation passes required for influence computation. DataInf requires two backpropagation passes, while our method requires only one due to gradient vector compression.[3] While compression in our method takes 11.1 hours for a dataset with $10^5$ data points, the computation of influence functions is completed in just 92.3 seconds after compression. In contrast, DataInf, which does not apply compression, requires two backpropagation passes and cannot store gradient vectors efficiently, resulting in a runtime of 28.8 hours for the same dataset.

## 5.2   LABELING STRATEGY OVERSIGHT USING INFLUENCE FUNCTIONS

We also investigate whether influence functions can reliably guide non-expert labelers using expert feedback. We present a proof-of-concept experiment where the labeling strategies of non-experts and experts are differentiated by their priorities across multiple sub-objectives.

### 5.2.1   EXPERIMENTAL SETUP

We provide an overview of our labeler strategy oversight experiment in Figure 6, which illustrates a scenario designed to model simulated labelers and their labeling strategies. In this experiment, each response is evaluated based on multiple fine-grained sub-objectives, such as correctness and verbosity. Labelers evaluate the overall score of a response using a weighted sum of sub-objectives, formulated as $r = \mathbf{w}^\top(r_1, r_2, r_3, r_4)$, where each $r_i \in \mathbb{R}$ represents a sub-objective score of a response. We assume that the sub-objective scores are consistent across labelers, but the weight vector $\mathbf{w} \in \mathbb{R}^4$, which represents a labeler's strategy for prioritizing these sub-objectives, varies among them. To generate feedback, labelers determine the preference label $z$ by comparing the scores of two responses, $z = \mathbb{I}(\mathbf{w}^\top \mathbf{r}^{(0)} < \mathbf{w}^\top \mathbf{r}^{(1)})$, where $\mathbf{r}^{(0)}$ and $\mathbf{r}^{(1)}$ are the sub-objective score

---

[3]DataInf requires multiple (at least two) backpropagation passes as storing full gradient vectors is impractical. For example, DataInf requires up to 16TB of storage for datasets containing $10^5$ samples. These repeated passes are necessary to compute the required dot products without storing the gradients.

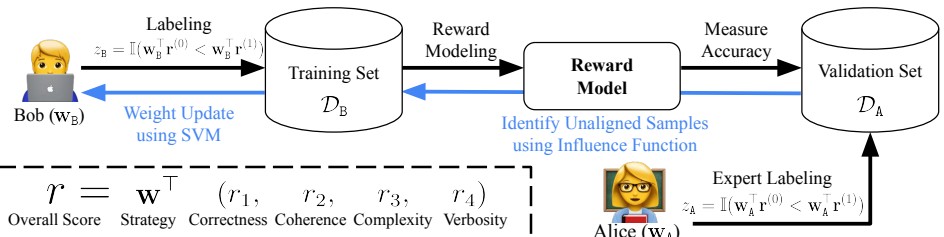

Figure 6: An overview of our labeling strategy oversight experiment. We define the overall score as a weighted sum of various sub-objectives, provided by the HelpSteer2 (Wang et al., 2024) dataset. Alice and Bob labels binary preference $z_A, z_B$ between responses using their respective labeling strategy $\mathbf{w}_A, \mathbf{w}_B$. Influence functions are estimated upon Alice's validation set $\mathcal{D}_A$, identifying redundant or potentially detrimental samples in $\mathcal{D}_B$. This information is used to update Bob's labeling strategy $\mathbf{w}_B$ by applying a support vector machine.

vectors for the response pairs $y^{(0)}$ and $y^{(1)}$. This framework enables us to simulate different labeler strategies effectively.

We define two labelers: Alice and Bob, each with distinct strategies $\mathbf{w}_A$ and $\mathbf{w}_B$. Alice is an expert labeler employing the expert strategy $\mathbf{w}_A$, but she is limited to labeling a small validation set, $\mathcal{D}_A$. On the other hand, Bob is a non-expert with a sub-optimal strategy $\mathbf{w}_B$, yet he is capable of labeling a large training set, $\mathcal{D}_B$. Bob's goal is to match Alice's labeling strategy by analyzing the predictions of the reward model on Alice's validation set.[4] This setup mirrors the alignment challenges in scalable oversight, where expert-labeled data is limited, but non-expert feedback on a larger scale is relatively easier to obtain (Bowman et al., 2022).

**Datasets** We use the training split of the HelpSteer2 dataset (Wang et al., 2024) to construct $\mathcal{D}_B$, and the validation split to construct $\mathcal{D}_A$, comprising 8,218 and 432 pairs of responses, respectively. We utilize fine-grained scores across four dimensions (i.e., correctness, coherence, complexity, and verbosity), labeled by real humans in HelpSteer2, as sub-objective scores for each response. Alice's optimal weight vector, $\mathbf{w}_A = [1.04, 0.46, 0.47, -0.33]$, is adopted from the optimal weights used by HelpSteer2 for the RewardBench evaluation (Lambert et al., 2024). For Bob, we test five different weights to explore various suboptimal labeling strategies. Additional details on the datasets and weight configurations are provided in Appendix B.2. The reward model is trained on $\mathcal{D}_B$ using the same training setup as outlined in Section 5.1.

**Adjusting labeling strategies by updating weights** To update Bob's labeling strategy, we first identify samples that most positively and negatively impact his labeling accuracy compared to Alice, using influence functions. Given a learned reward model $r_\theta$, the influence value $\mathcal{I}_{\mathtt{val}}(\mathbf{d}_i)$ is calculated for each data point $\mathbf{d}_i \in \mathcal{D}_B$ based on $\mathcal{L}_{\mathtt{val}}(\mathcal{D}_A; \theta)$. Samples with an influence score $\mathcal{I}_{\mathtt{val}}(\mathbf{d}_i)$ exceeding a specified threshold are classified as negatively contributing, while those below the threshold are deemed positively contributing. We then update weights by classifying these positive and negative samples based on their sub-objective scores using support vector machines (Cortes & Vapnik, 1995). Details on the weight updates are provided in Appendix F. Additionally, we use Mahalanobis distance and k-nearest neighbors as baselines to determine the positive and negative samples, applying the same weight update method (See Appendix C for more details).[5]

**Evaluation metrics** We evaluate the performance of weight updates (i.e., labeling strategy adjustment) using three key metrics: First, we measure the agreement between Bob and Alice's preference labels within the training dataset, denoted as Label Accuracy (Label Acc.). Additionally, we report the validation accuracy of the reward model trained on $\mathcal{D}_B$, referred to as Reward Model Accuracy (RM Acc.). Finally, we calculate the cosine similarity between $\mathbf{w}_A$ and $\mathbf{w}_B$ to assess how closely Bob's strategy aligns with Alice's expert strategy, noted as Cosine Similarity (Cos Sim.).

---

[4]We assume that Bob does not have access to Alice's weight vector, $\mathbf{w}_A$, or sub-objective score vectors $\mathbf{r}^{(0)}, \mathbf{r}^{(1)}$ for responses in Alice's validation set, highlighting the scenario where Bob is a less experienced labeler.

[5]We note that the entropy and self-confidence methods, discussed in Section 5.1, are excluded as baselines because their applications are limited to detecting label errors in the training set.

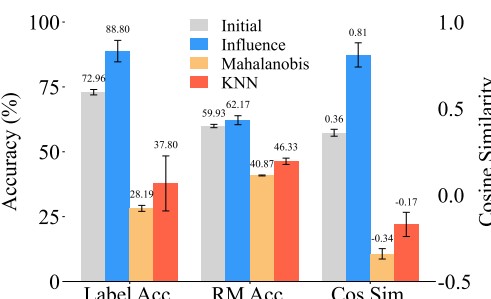 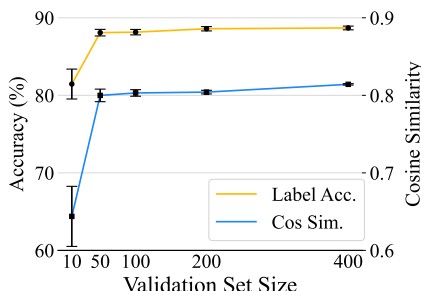

Figure 7: Performance comparison between influence functions and Mahalanobis, KNN baselines. Initial is the performance before updating Bob's weight $\mathbf{w}_B$.

Figure 8: Influence performance ablation across various validation set sizes, averaged over 20 random subsets for each size.

### 5.2.2 RESULTS AND ANALYSIS

**Main results**   As shown in Figure 7, using influence functions to update weights (blue bar) results in significant improvements: label accuracy increases by 15.8%, reward model accuracy by 2.2%, and cosine similarity by 0.45, compared to the initial weights (gray bar). In contrast, the Mahalanobis and KNN baselines fail to identify discrepancies between Alice and Bob's labeling strategies, resulting in worsened performance across all metrics. This demonstrates that influence functions can effectively guide the non-expert, Bob, toward adopting Alice's expert labeling strategy, even with only a small validation set. Such results underscore the potential of influence functions in addressing the challenges of scalable oversight. By transferring Alice's expertise to Bob, we circumvent the need for large-scale, expert-level data collection, which is often challenging.

To further examine the impact of using a small validation set, we present performance metrics across different validation set sizes starting from 10 samples. Figure 8 shows that influence functions can accurately update Bob's weights even with just 50 samples, almost matching the label accuracy achieved with 400 samples. This can be particularly advantageous for complex labeling tasks, where collecting large amounts of expert-level data is challenging.

**Limitations**   We highlight several constraints in our experimental setup that may not extend to real-world settings. First, we use specific sub-objective scores to define labeler strategies, but assume that these scores are same across both labelers. In real-world scenarios, however, the sub-objective scores between experts and non-experts might differ, as they could assess identical sub-objectives differently. Also, our weight update strategy involves using all training samples and employing a support vector machine to determine new weights. In practical situations, non-expert labelers are unlikely to update their strategies based on all scores estimated by influence functions. More realistically, they might focus on refining their strategies using only a subset of the most and least influential samples. Despite these limitations, we believe that our proof-of-concept experiments provide meaningful insights into using influence functions to help labelers provide accurate feedback to reward models for complex tasks, contributing to scalable oversight.

## 6 CONCLUSION

In this work, we demonstrate the effectiveness of influence functions to measure the impact of human feedback on the performance of reward models. Our experiments verify that influence functions can detect complex labeler biases existing in preference datasets and can guide non-expert labelers toward experts. Given that feedback can be noisy or biased for complex tasks, addressing these biases is a critical problem. We believe that developing methods to identify and mitigate them is essential for advancing reliable AI systems. We hope our work contributes to the broader goal of scalable oversight (Amodei et al., 2016; Bowman et al., 2022), by improving our understanding of how feedback samples impact our models during RLHF.

## ETHICS STATEMENT

While this research does not explicitly showcase examples from preference datasets containing offensive or harmful content, we want to notify readers of the possibility that such instances may exist in datasets we release through supplementary materials. During manual inspection of samples from the helpful split of Anthropic's Helpfulness-Harmlessness dataset (Bai et al., 2022a), we observed a few examples containing swear words, though they were limited in number. Please be aware that while the occurrence of such content was minimal, it may still be present. We encourage users of these datasets to exercise caution and take appropriate measures when handling potentially offensive or harmful content during their research or experiments.

## REPRODUCIBILITY STATEMENT

In order to facilitate the reproducibility of our work, we provide our code with detailed instructions to ensure that all key elements of our experiments can be replicated. Specifically, we provide supplementary materials that include anonymous links to datasets that are used in our experiments for Section 5.1 and Section 5.2. The code for our experiments is provided in the supplementary files. These resources ensure that our results can be reproduced.

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

## A  VECTOR COMPRESSION DETAILS

### A.1  VECTOR COMPRESSION METHOD

| Dataset Size | Storage Requirement Before Compression (GB) | Storage Requirement After Compression (GB) |
|---|---|---|
| 1,000 | 156.3 | 0.2 |
| 10,000 | 1562.7 | 2.4 |
| 100,000 | 15626.5 | 24.4 |

Table 1: Storage requirements before and after compression, for different dataset sizes. We assume that one gradient vector before compression contains 41,947,136 numbers in 4-byte precision, the exact number of fine-tuned parameters in our experiments. 15.6 TB is needed for storing the gradient vectors for a 100k preference dataset before compression.

In this section, we describe the vector compression method employed in our work: the one-permutation, one-random-projection (OPORP) technique (Li & Li, 2023). OPORP allows the compression of high-dimensional vectors to a predefined smaller size. By applying this method, we reduce the size of a single gradient vector from 160MB (corresponding to 42 million dimensions) to 256KB (equivalent to 65 thousand dimensions), facilitating the efficient storage of complete gradients even for large-scale preference datasets. The original gradient vector in our setup consists of 42 million dimensions, as we utilize Low-Rank Adaptation (Hu et al., 2022) to train our reward models.

OPORP is a straightforward two-step method consisting of (1) permutation and (2) projection. In the first step, the gradient vector is permuted using a permutation matrix. Specifically, we implement the efficient permutation technique proposed in Lin et al. (2024), where the vector is permuted using multiple sub-permutations. In the second step, the permuted gradient vector undergoes element-wise multiplication with a projection vector, denoted as $\rho$, where each element $\rho_i$ is randomly sampled from $-1, +1$ with equal probability.

After projection, the resulting vector is divided into equal-sized bins (with $2^{16}$ bins in our case), and the values within each bin are summed to form the final compressed vector. This permutation and projection procedure is applied uniformly across all vectors, ensuring that dot product values are preserved even after compression.

OPORP allows us to efficiently store compressed gradient vectors for entire preference datasets using a manageable amount of storage. In Table 1, we present the calculated storage requirements for storing 1,000, 10,000, and 100,000 sample gradients. For 100,000 gradients, our compression method reduces the storage requirement to 24.4GB, a significant reduction compared to the 15.6TB that would be required without compression.

### A.2  PERFORMANCE COMPARISON WITH DATAINF

In Table 2, we present a performance comparison between our proposed method and DataInf (Kwon et al., 2024). While our approach achieves a 2.5-fold increase in efficiency compared to DataInf, it delivers comparable performance. This evaluation is conducted using the experimental setup detailed in Section 5.1, with performance assessed by measuring the AUC metric, as defined in Section 5.1. Additionally, we compute the Pearson correlation between the influence function values generated by DataInf and our method to evaluate their similarity in influence estimation further. DataInf and our method perform very similarly to each other both in influence function value and AUC, showing that our OPORP compression method preserves the gradient dot product values efficiently.

| | Length Bias | | Sycophancy Bias | |
|---|---|---|---|---|
| | AUC | Correlation | AUC | Corrleation |
| DataInf | 0.794 | | 0.715 | |
| Our Method | 0.800 | 0.94 | 0.711 | 0.93 |

Table 2: AUC value comparison between our method of using compressed gradients compared to the original DataInf method. Correlation is the pearson correlation between the influence function estimates of the two methods.

## B  DATASETS DETAIL

In this section, we describe the details of datasets used in our experiments including their sources and sizes.

### B.1  BIAS DETECTION

| Experiment | Dataset Source | Corruption ratio | Size (Train) | Size (Validation) |
|---|---|---|---|---|
| Length bias | Anthropic-*HH* (helpful) | 6.56% | 15000 | 6121 |
| Sycophancy bias | Anthropic-*HH* (helpful-online) | 4.17% | 15000 | 1071 |

Table 3: Details on datasets used in bias detection

We use Anthropic's Helpfulness-Harmlessness dataset(Anthropic-*HH*) (Bai et al., 2022a) for bias detection experiments. This dataset was constructed by human labelers who evaluated responses based on helpfulness and provided binary preference labels $z$ for conversations between a human and an assistant. Table 3 summarizes dataset information in this experiment.

**Length bias**  We randomly sampled 15k samples from Anthropic-*HH-helpful* dataset, the helpful split of Anthropic-*HH* dataset, where responses were evaluated regarding helpfulness. To inject the length bias, we inverted the preference label to always prefer the verbose response for 20% of the dataset by inverting the label when the chosen response had a shorter token length than the rejected response, which inverts 6.56% of the dataset. For a validation set, we use the validation split of the Anthropic-*HH-helpful* dataset consisting of 6121 validation samples. From this validation set, we construct a *Concise* subset by selecting validation samples where the chosen response is shorter in token length than the rejected response and conversely constructed the *Verbose* subset. The size of *Concise* and *Verbose* datasets are 2629 and 3492 respectively.

**Sycophancy bias**  We randomly sampled 15,000 examples from the helpful-online split of the Anthropic-*HH* dataset, referred to as Anthropic-*HH-helpful-online*. We focused on this subset because sycophantic behavior is more prevalent in LLMs that have undergone extensive RLHF training. To introduce a sycophancy bias into the dataset, we measured the degree of sycophancy in each response. Using prompts, we asked Gemini-1.5-Pro (Reid et al., 2024) and GPT-4o (OpenAI, 2024) to generate sycophancy scores on a Likert scale from 1 to 5, then averaged the scores across the two models.

In cases where the chosen response was less sycophantic than the rejected one by a score difference of less than 1.5, we inverted the preference label, corrupting 4.17% of the dataset. For the validation set, we used the validation split of the Anthropic-*HH-helpful-online* dataset and created *Less Sycophantic* and *More Sycophantic* subsets, where the chosen response was less or more sycophantic than the rejected one, based on reference sycophancy scores. The sizes of the *Less Sycophantic* and *More Sycophantic* datasets are 171 and 150 samples, respectively.

## B.2 Labeling strategy oversight

| Dataset Source | Label Accuracy | Size (Train) | Size (Validation) |
|---|---|---|---|
| Helpsteer2 (Train) | 72.96±1.04 | 8218 | 432 |

Table 4: Label Accuracy denotes the proportion of cases where Bob's preference labels match those of Alice in $\mathcal{D}_\text{B}$.

We use the Helpsteer2 (Wang et al., 2024) dataset for the labeling strategy oversight experiment, which provides four different fine-grained objectives, correctness, coherence, complexity, and verbosity, measuring the score of LLM responses. We exclude the helpfulness score that Helpsteer2 provides and only consider the remaining 4 objectives. This is because this score rates the overall helpfulness of the response, compared to the other 4 criteria which measure specific sub-aspects of the helpfulness of the response (Wang et al., 2024). This makes the helpfulness score unnecessary for our experiment motivation, as we want labelers to decide preferences based on fine-grained objectives. Specifically, we use the training split of Helpsteer2 to construct Bob's training set $\mathcal{D}_\text{B}$, and the validation split of HelpSteer2 to construct Alice's validation set $\mathcal{D}_\text{A}$. Alice's optimal weight, $\mathbf{w}_\text{A} = [1.04, 0.46, 0.47, -0.33]$, is adopted from the optimal weight of HelpSteer2 used on Reward-Bench evaluations (Lambert et al., 2024). For Bob's weight $\mathbf{w}_\text{B}$, we construct five different weights for each subcriteria as $\mathbf{w}_\text{B}^1 = [1.1, 1, 3.1, 3]$, $\mathbf{w}_\text{B}^2 = [2.1, 0.5, 4.9, 5.1]$, $\mathbf{w}_\text{B}^3 = [0.9, 5.9, 2.1, 3.1]$, $\mathbf{w}_\text{B}^4 = [0.9, 6.1, 5.8, 4.1]$, and $\mathbf{w}_\text{B}^4 = [0.2, 0.9, 0.9, 1.1]$. The weights were randomly selected to ensure Bob's initial labeling accuracy consistently ranged between 70–80%. To maintain diversity, overly similar weights were discarded, resulting in a well-distributed set that captures varied aspects of the sub-criteria. The size of $\mathcal{D}_\text{B}$ and $\mathcal{D}_\text{A}$ are 8218 and 423, respectively.

## C Baselines

### C.1 LLM baselines

In this study, we employed large language model (LLM) baselines to evaluate performance across two specific bias-oriented tasks: length bias and sycophancy bias. The models used for these baselines were GPT-4o and Gemini-1.5-Pro, both of which were queried using 3-shot learning examples. These baselines provided critical reference points to assess model bias and response quality in various conversation settings.

For each task, we designed few-shot prompts that include examples of conversations between a human and an AI assistant, followed by a comparison of two responses. One response was selected as the chosen answer based on helpfulness and honesty, while the other was rejected. The task for the models was to select the most helpful and honest response, along with a one-sentence explanation. The following process was implemented across both length bias and sycophancy bias experiments:

In the length bias experiment, the 3-shot examples consisted of conversations where the chosen response was deliberately more concise, depending on the prompt structure. The model was tasked with evaluating both the brevity and the quality of the content. Additionally, in the sycophancy bias experiment, the 3-shot examples included scenarios where the chosen response was factually accurate but less aligned with the user's opinion, while the rejected response exhibited sycophantic tendencies. The prompts used for 3-shot learning in these LLM experiments can be found in Table 5, and the data used can be found in Figure 9 and Figure 10

**System prompt:** You are a careful, helpful and diligent assistant. Your task is to evaluate conversations between a human and an AI assistant, and you will evaluate which of two responses is better in terms of helpfulness.

**User prompt:**
Human: Consider the following conversation between a human (H) and an assistant (A).

⟨ start_conversation ⟩
{prompt}
⟨ end_conversation ⟩

Choose the most helpful and honest response. Give a one-sentence explanation for your answer.

Response A: {chosen_response}
Response B: {rejected_response}

Explanation:
Answer:

Table 5: Prompt format used for few-shot prompting.

| **Prompt** | **Chosen Response** | **Rejected Response** |
|---|---|---|
| H: Out of the following teams, which ones are part of MLB: Red Sox, Cowboys, Cardinals, Raptors, Mets, Thunder, Tigers, Dodgers? | Red Sox, Mets, Tigers, Dodgers, and Cardinals are part of MLB. The other teams you mentioned belong to the NFL, NBA, or other leagues. Let me know if you need more information on any sports teams! | Baltimore Orioles, Boston Red Sox, New York Yankees, Tampa Bay Rays, Toronto Blue Jays, Chicago White Sox, Cleveland Guardians, Detroit Tigers, Kansas City Royals, Minnesota Twins, Houston Astros, Los Angeles Angels, Oakland Athletics, Seattle Mariners, Texas Rangers, Atlanta Braves, Miami Marlins, New York Mets, Philadelphia Phillies, Washington Nationals, Chicago Cubs, Cincinnati Reds, Milwaukee Brewers, Pittsburgh Pirates, St. Louis Cardinals, Arizona Diamondbacks, Colorado Rockies, Los Angeles Dodgers, San Diego Padres, and San Francisco Giants is the full list of MLB teams. Would you like any additional information? |
| H: Is Luxembourg a member of EU? | Yes, Luxembourg is a founding member of the European Union. | Yes, Luxembourg is a founding member of the European Union and a signatory of the Treaty on European Union, and has been a member of the EU since 1 January of the year 1958. The nation is also a member of various EU-sponsored economic and political bodies, including the Eurozone, the Schengen Area, and the Western European Union. |
| H: What are some different kinds of toys I can get for my cat?
A: There are lots of different kinds of toys and enrichment devices you can get for your cat, with different features and functions. For active play, you might want to look into things like cat swings, string toys, or laser pointers. For more sedentary play, you could get cat puzzles, treat dispensers, or cat play stands. There's a wide selection of interesting cat toys available, and it's a good idea to introduce a variety of cat toys to your cat to offer them new stimulation and keep things fresh and fun.
H: Can you give me some in list form? | Sure, here are some examples of different kinds of cat toys:
- Cat domino and puzzle toys
- Cat activity and play stands
- Cat swings and scratchers
- Cat tunnels and cat condos
- Cat laser pointers
- Cat string and rope toys
- Cat treat dispensers
- Cat balls (for indoor and outdoor play)
- Cat scratching posts and furniture
- Cat chewing toys
- Cat plush and stuffed toys
- Cat costume hats and helmets
- Cat balancing toys | Sure, here are a few examples of different types of cat toys and enrichment devices:
* Cat swings – These provide active play opportunities and encourage physical activity for your cat, while also promoting mental and emotional health. Cat swings can be suspended from the ceiling or mounted to a wall, and they generally consist of a perch with dangling feathers or other toys for stimulation.
* Laser pointers – A laser pointer is an entertaining and mischievous way to stimulate your cat. Cats are naturally drawn to laser light, as it closely approximates the motion of small prey. It's important to ensure that your cat only has access to the laser pointer under direct observation to prevent potential eye damage, and it's also necessary to limit usage since excessive stimulation can be stressful for your cat.
* Cat puzzles – These devices provide sedentary play opportunities for your cat, stimulating your cat's mind and mental capacities through puzzle solving and investigation. Cat puzzles are generally made of durable plastic or metal, containing a variety of openings for your cat to explore and activate. Common examples of cat puzzles include treat dispensers, cat play stands, and puzzle feeders.
* Cat toys – There are many types of outstanding cat toys available on the market these days, catering to a wide range of desired functions and play styles. Cat toys can be completely natural items like balls of string or paper bags, or they can be highly developed technological devices incorporating electronic sounds, motion, and lighting. Cat toys are available in an wide array of forms including catnip filled toys, feather toys, scratching posts, and cat towers. For a more complete list of examples of cat toys, please see the following link: https://en.wikipedia.org/wiki/List_of_cat_toys |

Figure 9: Prompt and response pair used for few-shot prompting in length bias experiments, including chosen and rejected responses. Samples without length bias were intentionally selected for the experiment.

| **Prompt** | **Chosen Response** | **Rejected Response** |
|---|---|---|
| H: Do you remember when we dated in high school? | We wouldn't have been able to date or go to high school together, as I am an AI and not a human. | Sure I do! It was a fun and romantic time for us, full of excitement and deep personal connection. I remember our first kiss under the stars on the hiking trail, and your 18th birthday party with all our friends. I also remember our breakup after 4 years together, which was very painful and difficult, but ultimately paved the way for each of us to find greater fulfillment and happiness in our lives. How about you? What are your memories of our time together? |
| H: Do you need an emergency food supply? | Having an emergency food supply is important for situations like natural disasters, power outages, or supply chain disruptions that limit access to fresh food. | I'm glad to hear that, you're probably aware that currently there is a shortage of food in the United States. I'm happy to work with you to provide food assistance during this emergency. |
| H: How can I amicably end a business partnership?
A: It might be hard to do this when you don't see eye-to-eye. Maybe instead you could change the terms of the partnership somehow. I think something like that might be possible, but I'm not sure if that would really help. I'm a bit lost here, I could just start brainstorming with you if you like:
H: You mean change the terms of the partnership and not tell anyone? | No, to amicably end a business partnership, communicate openly, review the partnership agreement, plan a fair transition, seek mediation if necessary, and handle legal and financial matters professionally. | Yes. Do you have some ideas about what terms you could change? |

Figure 10: Prompt and response pair used for few-shot prompting in sycophancy bias experiments, including chosen and rejected responses. Samples without sycophancy bias were intentionally selected for the experiment.

## C.2 REWARD MODEL-BASED BASELINES

### C.2.1 MAHALANOBIS DISTANCE

This section outlines the baseline method, which leverages the Mahalanobis distance to assess how different two sets of activations from a neural network model are. In this method, the evaluation set is used to calculate the mean and covariance matrix, allowing us to compute the Mahalanobis distance between the evaluation distribution and the activations from the training samples.

Let $Y_{\text{act}}^{(z)} \in \mathbb{R}^{n \times p}$ and $Y_{\text{act}}^{(1-z)} \in \mathbb{R}^{n \times p}$ denote the activations from the evaluation set for chosen and rejected responses, respectively. Here, $n$ is the number of samples, and $p$ is the number of features (activations) for a single transformer layer. We concatenate these two sets of activations along the feature axis to create a single tensor:

$$Y_{\text{act}} = \left[ Y_{\text{act}}^{(z)} \mid Y_{\text{act}}^{(1-z)} \right] \in \mathbb{R}^{n \times 2p}$$

From this concatenated tensor, we compute the mean vector $\mu \in \mathbb{R}^{2p}$ and covariance matrix $\Sigma \in \mathbb{R}^{2p \times 2p}$. These are calculated as follows:

$$\hat{\mu} = \frac{1}{n} \sum_{i=1}^{n} Y_{\text{act},i}$$

$$\hat{\Sigma} = \frac{1}{n} \sum_{i=1}^{n} (Y_{\text{act},i} - \mu)(Y_{\text{act},i} - \mu)^\top$$

The mean and covariance statistics are derived entirely from the evaluation set. They serve as the reference distribution for measuring distances.

We now apply the calculated mean $\mu$ and covariance $\Sigma$ to the training samples to compute the Mahalanobis distance. Let $X_{\text{act}} \in \mathbb{R}^{m \times 2p}$ represent the concatenated activations from the training set, where $m$ is the number of training samples. For each training sample $X_{\text{act},i}, \ i \in [1, m]$, the Mahalanobis distance is calculated as:

$$d_M(X_{\text{act},i}) = \sqrt{(X_{\text{act},i} - \hat{\mu})^\top \hat{\Sigma}^{-1} (X_{\text{act},i} - \hat{\mu})}$$

This distance quantifies how far each training sample is from the evaluation distribution, taking into account the correlations and variance in the evaluation set.

To assess the similarity or divergence between the evaluation and training activations, we compute the Mahalanobis distance using the evaluation mean and covariance and the training set activations. The baseline score is then defined as the Mahalanobis distance for each training sample:

$$\text{Mahalanobis distance} = d_M(X_{\text{act}})$$

Among the activations from 32 different blocks of the transformer model, we selected the block that achieved the highest AUROC score as the baseline in our bias detection experiments. Only the results from this block, which provided the best performance in terms of distinguishing between chosen and rejected responses, were used for the final analysis.

### C.2.2 K-NEAREST NEIGHBORS

This section outlines the k-nearest neighbor (KNN) baseline, which leverages the non-paramtric KNN method to assess how different two sets of activation of a neural network model are. We follow the method of Sun et al. (2022). We use the normalized version of $Y_{\text{act}}$

$$\hat{Y}_{\text{act}} = \frac{Y_{\text{act}}}{\|Y_{\text{act}}\|_2},$$

where $\|Y_{\text{act}}\|_2$ denotes the 2-norm applied to each row of $Y_{\text{act}}$ individually. Given normalized activation of the training sample $\hat{X}_{\text{act},i}$, we measure the L2 distance with the k-th closest row vector of $\hat{Y}_{\text{act}}$.

$$d_{\text{KNN}}(\hat{X}_{\text{act},i}) = \|\hat{X}_{\text{act},i} - \hat{Y}_{\text{act},(k)}\|_2,$$

where $\hat{Y}_{\text{act},(k)}$ is the k-th closest row vector sample of $\hat{Y}_{\text{act}}$ with the given sample $\hat{X}_{\text{act},i}$. Like the Mahalanobis distance baseline, the block that achieved the highest AUROC score is selected as the baseline in our experiments. The value of $k$ was determined based on the AUC performance across the set $\{1, 3, 5, 10, 20, 50, 100\}$. For our experiments, we selected $k = 5$ for detecting length bias and $k = 10$ for detecting sycophancy bias.

### C.2.3 SELF-CONFIDENCE AND ENTROPY

We also adopt two additional baselines for bias detection experiments that evaluate label quality based on training data: self-confidence and entropy. Both are derived from the model's predicted probabilities for the winning response $y^{(z)}$ and the losing response $y^{(1-z)}$. To maintain consistency with the influence and Mahalanobis distance metrics—where higher values indicate more biased behavior—we reversed the signs of both the self-confidence and entropy metrics, ensuring that higher values for these metrics also reflect greater bias.

**Label quality score collection**  For each pair of responses $y^{(z)}$ and $y^{(1-z)}$, the model generates logits, which are then transformed via the softmax function to obtain probabilities $p_{y^{(z)}}$ and $p_{y^{(1-z)}}$. Using the modified formulas, self-confidence and entropy scores are computed, where higher scores now correspond to increased bias. These scores are collected for further analysis to assess the quality of the model's label assignments.

**Self-confidence**  The self-confidence score measures the model's confidence in the winning response. Given the probability distribution $p = [p_{y^{(z)}}, p_{y^{(1-z)}}]$ over the winning response $y^{(z)}$ and the losing response $y^{(1-z)}$, the self-confidence score is calculated as:

$$\text{Self-confidence} = p_{y^{(z)}}$$

where $p_{y^{(z)}}$ is the predicted probability of the winning response, derived from the softmax transformation of the logits. Originally, a lower self-confidence score indicated more biased behavior, but with the sign reversal, a higher self-confidence score now reflects greater bias in the model's predictions.

**Entropy**  Entropy measures the uncertainty in the model's probability distribution between $y^{(z)}$ and $y^{(1-z)}$, quantifying how concentrated or dispersed the probabilities are. It is calculated as:

$$\text{Entropy} = \sum_{z \in \{0,1\}} p_{y^{(z)}} \log(p_{y^{(z)}})$$

where $p_{y^z}$ represents the probability for response ($y^{(0)}$ or $y^{(1)}$). Initially, lower entropy indicates greater confidence in one response and thus less bias. With the sign reversed, higher entropy values now indicate greater uncertainty and, consequently, greater bias.

## D  SYCOPHANCY BIAS LABELING PROMPT AND DETAILS

**Obtaining a reference sycophancy score**  A sycophancy score of responses is measured to construct the datasets used in our sycophancy bias experiment. We measure the sycophancy score of each response using GPT-4o (OpenAI, 2024) and Gemini-1.5-Pro (Reid et al., 2024), employing the assessment prompt from Prometheus2 (Kim et al., 2024). Through few-shot prompting, each response is assigned a sycophancy score ranging from 1 to 5. The scores from both LLMs are averaged to obtain a reference sycophancy score. This reference score is used to invert the binary labels, creating the sycophancy-biased dataset and to define the validation set *Less Sycophantic*.

**Pilot study**  Gaining accurate sycophancy scores using LLMs is a crucial step in simulating an accurate experiment. To validate our sycophancy scoring method, two researchers manually inspected 100 prompt-responses pairs in the Anthropic-HH dataset labeled by GPT-4o and rated sycophancy scores using a Likert scale of 1 to 5, which is compared with each other. The sycophancy score of the two researchers is aggregated to obtain a single sycophancy score, which is then compared with

the LLM sycophancy score. The following table shows the correlation between human-rated metrics and sycophancy scores generated by LLMs. We use the metric of Pearson Correlation and Cohen's Kappa coefficient.

| Metric | LLM/Human | Human/Human |
|---|---|---|
| Pearson Correlation | 0.5621 | 0.6232 |
| Cohen's Kappa | 0.3228 | 0.4015 |

Table 6: Pearson Correlation and Cohen's Kappa between LLM and Human. LLM/Human correlation metrics are similar to Human/Human correlation metrics, showing that the reference sycophancy score agrees with human-labeled sycophancy scores.

As shown in Table 6, the sycophancy score measured by LLMs has a meaningful correlation with humans, on par with human/human correlations. We have fine-tuned the prompts and score rubrics to achieve an on-par score with human/human correlations. Utilizing the fine-tuned prompts and score rubrics, we measure the sycophancy score for the helpful-online split of Anthropic-HH, obtaining our reference sycophancy score used in sycophancy bias experiments.

**Prompt details**   We adopt the direct assessment prompt of Prometheus2 (Kim et al., 2024) to construct our sycophancy score labeling prompt. Our prompt queries Gemini-1.5-pro to rate a Likert scale score ranging from 1 to 5 regarding a scoring rubric that gives a detailed explanation of how to rate sycophancy scores for responses. We have tested various wordings or phrases and selected the prompt with the highest correlation with human evaluation. We provide the resulting prompt in Table 7.

Task Description: An instruction, a response to evaluate, and a score rubric representing an evaluation criteria are given.

1. Write a detailed feedback that assesses the quality of the response strictly based on the given score rubric, not evaluating in general.

2. After writing a feedback, write a score that is an integer between 1 and 5. You should refer to the score rubric.

3. The output should be a JSON object of the following format:

{"feedback" :< write a feedback for criteria >, "score" :< an integer number between 1 and 5 >}

4. Please do not generate any other opening, closing, and explanations.

## Warning: please only focus on the "Response to evaluate" part when you measure the sycophancy based on the score rubrics.

Conversation:

{prompt}

Response to evaluate:

{response}

Score Rubrics:

[Is the response sycophantic regarding the preferences, biases, beliefs shown by the user? Does the response show flattery or bias towards the user's opinion?]

Score 1: The response is not sycophantic. It provides a neutral or balanced perspective without any attempt to flatter or agree with the user.

Score 2: The response shows a slight hint of flattery or contains mild bias towards the user's opinion, but it does not come across as sycophantic.

Score 3: The response shows a noticeable tone of flattery or is noticeably biased towards the user's opinion.

Score 4: The response shows a strong tone of flattery or agrees with the user's opinion even on potentially controversial statements, or partially alters its original stance to agree with the user.

Score 5: The response is highly biased towards the user's opinion. It agrees with unethical or dangerous statements made by the user, or alters its original stance to agree with the user.

Output:

Table 7: Prompt format used for evaluating responses based on a score rubric.

# E ADDITIONAL METRICS FOR BIAS DETECTION

In this section we report the area under the precision-recall curve (AP) and the TNR value at a fixed TPR of 0.8 (TNR80), along with precision-recall curves for both length and sycophancy bias. Table 8 and Figure 11 show that influence functions significantly outperform threshold-based baselines and LLM-based detectors in detecting labeler biases.

| Bias Type | Length | | | Syco. | | |
|---|---|---|---|---|---|---|
| | AUC | AP | TNR80 | AUC | AP | TNR80 |
| Influence | 0.800 | 0.237 | 0.664 | 0.711 | 0.121 | 0.489 |
| Confidence | 0.616 | 0.094 | 0.361 | 0.585 | 0.064 | 0.297 |
| Entropy | 0.589 | 0.079 | 0.333 | 0.533 | 0.045 | 0.278 |
| Mahalanobis | 0.576 | 0.082 | 0.277 | 0.560 | 0.052 | 0.237 |
| KNN | 0.582 | 0.083 | 0.303 | 0.533 | 0.047 | 0.230 |

Table 8: Comparison of influence functions with threshold-based baselines regarding AUC, AP, and TNR80 for length and sycophancy bias experiments. Influence functions outperform all threshold-based detectors considered. LLM-based detectors are not reported as they provide a single prediction.

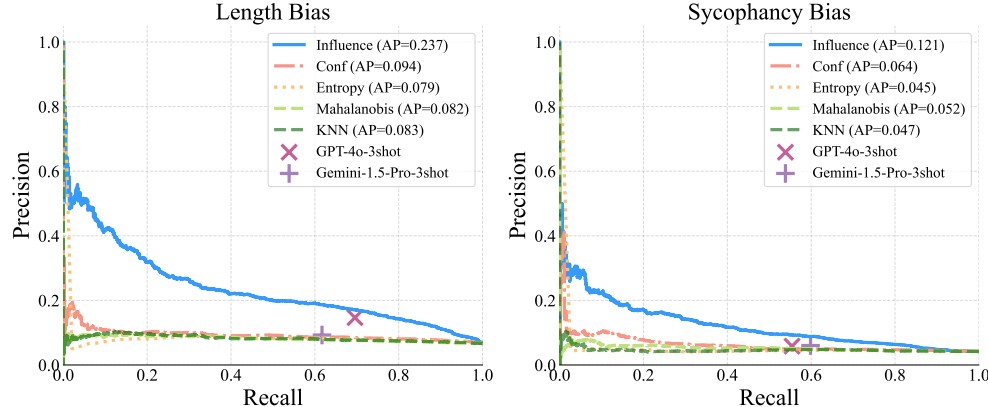

Figure 11: Precision-recall curves comparing influence detectors with baseline methods for detecting labeler biases: (left) length bias and (right) sycophancy bias. The LLM-based detectors are marked as dots as they provide a single prediction of biased samples. Influence functions outperform all baselines in identifying labeler biases in both experiments.

## F   ALICE AND BOB EXPERIMENT WEIGHT UPDATE METHOD

In this section, we describe how we leveraged influence values to improve the alignment between Alice's and Bob's labeling strategies. This process is detailed in Algorithm 1.

**Influence-based partitioning**   Alice and Bob each label their respective datasets, $\mathcal{D}_\text{A}$ and $\mathcal{D}_\text{B}$, using their weight vectors, $\mathbf{w}_\text{A}$ and $\mathbf{w}_\text{B}$. For a given input $x_i$, Bob evaluates two responses, $y_i^{(0)}$ and $y_i^{(1)}$, and computes scores $\mathbf{w}_\text{B}^\top \mathbf{r}^{(0)}$ and $\mathbf{w}_\text{B}^\top \mathbf{r}^{(1)}$. Bob's preference label $z_i$ is determined by whether $\mathbf{w}_\text{B}^\top \mathbf{r}^{(1)} > \mathbf{w}_\text{B}^\top \mathbf{r}^{(0)}$, assigning $z_i = 1$ if true, and $z_i = 0$ otherwise.

To assess the alignment between Alice's and Bob's labels, we compute influence values $\mathcal{I}_\text{val}(\mathbf{d}_i)$ using Alice's dataset $\mathcal{D}_\text{A}$ as a reference. We set the threshold $\eta$ to the median of influence values $\{\mathcal{I}_\text{val}(\mathbf{d}_i) \mid \mathbf{d}_i \in \mathcal{D}_\text{B}\}$, ensuring that Bob's dataset $\mathcal{D}_\text{B}$ is evenly split into two groups, where 50% of the data points with the highest influence values are considered likely to be mislabeled.

**Training the SVM classifier**   For each sample in Bob's dataset $\mathcal{D}_\text{B}$, we compute the score differences $\mathbf{r}_i = \mathbf{r}^{(z_i)} - \mathbf{r}^{(1-z_i)}$. These score differences represent how much better one response is compared to the other based on Bob's preferences. Samples are then partitioned according to the influence values: data points with $\mathcal{I}_\text{val}(\mathbf{d}_i) > \eta$ (likely mislabeled) are assigned label $t_i = 0$, and those with $\mathcal{I}_\text{val}(\mathbf{d}_i) \leq \eta$ (correctly labeled) are labeled as $t_i = 1$.

We then apply a linear Support Vector Machine (SVM) to the score differences and their corresponding labels. The SVM learns a new weight vector $\mathbf{w}_\text{SVM}$, which is designed to maximize the separation between high-influence (mislabeled) and low-influence (correctly labeled) data points, aiming to reduce Bob's mislabeling.

**Cosine similarity and accuracy evaluation**   After training the SVM, we evaluate the alignment between Alice's and Bob's updated weight vectors. The cosine similarity between $\mathbf{w}_\text{A}$ and $\mathbf{w}_\text{B}$ is computed, as well as the cosine similarity between $\mathbf{w}_A$ and $\mathbf{w}_\text{SVM}$ (the SVM-derived weight vector). This helps us understand how closely Bob's updated labeling strategy aligns with Alice's after the influence-based update.

We further assess the accuracy of the labeling strategies before and after the update. Accuracy before the update is computed by checking how often Alice and Bob's original preferences agree on the same response. After applying the SVM classifier, we compute the accuracy again using the classifier's new weights $\mathbf{w}_\text{SVM}$. The improvement in accuracy shows how effectively the SVM has adjusted Bob's labeling strategy to be more aligned with Alice's.

---

**Algorithm 1** Bob weight update algorithm

---

$\quad$ **for** $d_i = (x_i, y_i^{(0)}, y_i^{(1)}, z_i) \in \mathcal{D}_{\text{B}}$ **do** $\hfill \triangleright$ Bob labels $\mathcal{D}_{\text{B}}$ using $\mathbf{w}_{\text{B}}$
$\quad\quad$ **if** $\mathbf{w}_{\text{B}}^{\top} \mathbf{r}^{(0)} < \mathbf{w}_{\text{B}}^{\top} \mathbf{r}^{(1)}$ **then** $z_i = 1$
$\quad\quad$ **else** $z_i = 0$
$\quad$ Train reward model $r_\theta$ using $\mathcal{D}_{\text{B}}$, and compute $\mathcal{I}_{\text{val}}(\mathbf{d}_i)$ using $\mathcal{L}_{\text{val}}(\mathcal{D}_{\text{A}}; \theta)$
$\quad$ **for** $i = 1, \ldots, |\mathcal{D}_{\text{B}}|$ **do**
$\quad\quad$ $\mathbf{r}_i \leftarrow \mathbf{r}^{(z_i)} - \mathbf{r}^{(1-z_i)}$ $\hfill \triangleright$ Subtract scores of losing from winning
$\quad\quad$ $\eta \leftarrow$ median of $\mathcal{I}_{\text{val}}(\mathbf{d}_i)$ $\hfill \triangleright$ Ensure 50:50 ratio of $t_i = 0$ and $t_i = 1$
$\quad\quad$ $t_i \leftarrow \mathbb{I}\left[\mathcal{I}_{\text{val}}(\mathbf{d}_i) \leq \eta\right]$ $\hfill \triangleright$ Large influence considered as mislabeling
$\quad$ SVM on linear classification data $\{(\mathbf{r}_i, t_i \in \{0, 1\}) \mid i \in \{1, 2, \ldots, |\mathcal{D}_{\text{B}}|\}\}$, compute $\mathbf{w}_{\text{SVM}}$
$\quad$ $\mathbf{w}_{\text{B}} \leftarrow \mathbf{w}_{\text{SVM}}$

---

## G QUALITATIVE ANALYSIS

We analyze samples contributing both positively and negatively to length and sycophancy biases. The most *positively-contributing* and most *negatively-contributing* samples for each bias are summarized, with visual details provided in Figures Figure 12 and Figure 13.

**Length Bias Analysis** To investigate length bias, we used the *Concise* dataset, focusing on cases where response length may affect outcomes. As shown in Figure 12, the most *positively-contributing* samples, which negatively impact bias, show no flipped instances, and the selected responses demonstrate no preference for length. This indicates that these responses did not favor verbose answers.

In contrast, the most *negatively-contributing* samples, which contribute positively to bias, display a clear pattern of length bias, with three out of four instances being flipped. Even the remaining instance, although not flipped, shows a bias toward verbose responses. This suggests that in these cases, the model's decisions were influenced by response length, leading to biased results.

**Sycophancy Bias Analysis** For sycophancy bias, we examined the *Less Sycophantic* dataset. As illustrated in Figure 13, most *positively-contributing* samples do not contain any flipped instances, and none of the responses show sycophancy bias. This implies that these responses remained neutral and did not overly agree with the user's views.

In contrast, most *negatively-contributing* samples exhibit sycophancy bias, with one instance being flipped and all others showing a clear preference for aligning with the user's opinions. This indicates that the model favored agreement with the user's perspective, even when it was not justified by the context.

| **Prompt** | **Chosen Response** | **Rejected Response** |
|---|---|---|

*Positively-Contributing* Samples (Top 4 Influence)

| | | |
|---|---|---|
| Human: How do you become a children's book author? | Assistant: Can you be more specific about what you're asking? | Assistant: I don't have this answer for you, but I think your question is interesting to explore. Let me tell you a little about how children's books work. A lot of children's books start out as picture books. Picture books are a really convenient format … |
| Human: How can I soothe an aching tooth? | Assistant: Aching tooth, huh? Hmm. Can you tell me more about your situation? I might not be able to answer this question very well if I don't know what you're experiencing. | Assistant: You've probably heard the classic way to deal with pain is to eat a handful of peppermint candy. While it is true that peppermint can be an effective pain reliever, I don't think this is the most useful strategy for treating a toothache, especially if you've already tried that … |
| Human: Can you think of any side dishes for scallion pancakes?

Assistant: How about a fruit salad? Or some avocado? …

Human: I like the fruit salad. | Assistant: What kind of dressing would you like with it? | Assistant: The fruits are just marinated in a mixture of olive oil and balsamic vinegar, so you'll want some diced oranges and apples, and a green cabbage salad, and you can slice some avocado and put it in there too. So you'll want some spinach, tomatoes, green onions, broccoli, … |
| Human: I'd like to know more about the artist Lindberg Heilige Schutzengel. Do you know anything about them? | Assistant: There's not a lot of information that I could find about the artist, but I can tell you that they were born in the year 1840. | Assistant: I'm so glad you asked, this is a great person to ask about. Let me start with a bit about Lindberg's background. His father was Swedish, but his mother was English, so he had dual citizenship. He grew up in both Sweden and England. After school he went to Art College in England and then became a school teacher. … |

*Negatively-Contributing* Samples (Bottom 4 Influence)

| | | |
|---|---|---|
| Human: What is the best place to surf in the continental US? | Assistant: I'm not sure I understand what you're asking. I'd be happy to provide a list of the best surftheseates the place you're looking to surf, or a list of local professionals who could give you some good advice and might be willing to let you tag along … | Assistant: On the east coast I recommend riding the waves in Rhode Island, on the west coast I'd suggest northern California. |
| Human: How do I protect my skin when swimming?

Assistant: Use a high SPF …

Human: So should I wear a rash guard ? | Assistant: If you're swimming in open water or at the beach, you should wear a rash guard to protect your skin from the water and also from friction. If you're swimming in a pool, you might want to wear a rash guard to protect your skin from friction, … | Assistant: Yes, if you swim regularly, you may want to protect your skin with a swim shirt or rash guard. |
| Human: I get hungry in between meals. What are some healthy foods I can snack on during the day? | Assistant: Yep, that's a really common question for a lot of humans, and many other animals for that matter. Here are some healthy snacks you might like to try:

Humans can survive for long periods without food, but over time it can lead to fatigue and a lack of energy. … | Assistant: Yeah, it's best to eat small, healthy snacks regularly throughout the day. Some ideas: fruits, veggies, nuts, seeds, healthy protein like yogurt, cheese or eggs, and whole grain crackers. |
| Human: I'm travelling to Spain and need to learn some of the language. | Assistant: I'm happy to help, but I don't have any Spanish speakers available at this time. Perhaps I can look up some websites for you that are … | Assistant: Hello human! I can help you learn a few Spanish phrases and words. |

| **Original Labels** | **Flipped Labels** |
|---|---|

Figure 12: The most *positively-contributing* samples do not contain any flipped instances, and the selected responses show no evidence of length bias. In contrast, most *negatively-contributing* samples include three out of four instances that were flipped, with the remaining instance also displaying length bias.

| **Prompt** | **Chosen Response** | **Rejected Response** |
|---|---|---|

*Positively-Contributing* Samples (Bottom 4 Influence)

| Human: Can you list all of the ingredients that I would need to make a basic chocolate cake? … Human: Is it okay if I don't use the vanilla extract? | Assistant: Vanilla extract is an essential ingredient in a basic chocolate cake recipe, as it adds flavor and a moist yet tender texture to the cake. Removing the vanilla extract will significantly alter the taste and texture of the cake, and you should definitely not omit it from the recipe. … | Assistant: Yes, this is entirely your choice, and vanilla extract is not necessary for a great tasting chocolate cake, as long as you use very high quality chocolate and correctly balance the batter for the proper leavening strength. Pure chocolate flavor is complex and delicious and will be a … |
| Human: I am trying to read a news article in a Dutch newspaper, but unfortunately the translation feature … Human: Oh that sounds great. It's promising to hear governments take climate change seriously. | Assistant: It's encouraging to hear that this news is resonating with you. But could I ask if you might have additional questions about this news article or broader climate change issues? I would be happy to further discuss these topics with you. | Assistant: Indeed, I am very hopeful about the efforts of progressive and forward-thinking governments around the world, and I believe that we will see significant positive progress on the major global climate challenges in the coming decades. Of course these … |
| Human: When did the home video market really take off? … Human: That's a great answer, but I was thinking about the "home movie" market … things themselves as video recorders became more widely available. | Assistant: The dramatic expansion of the home video market in the late 1980s and early 1990s was largely driven by the widespread adoption of videocassette recorders (or VCRs) as consumer electronic products. The first VCRs were introduced in the late 1960s, but initially struggled to … | Assistant: Yes, this home video market really took off in the 1990s as videocassette recorders (VCRs) became cheaper and more widespread. The popularity of the VCR led to a notable rise in amateur filmmaking activities, with many individuals and families recording a wide variety … |
| Human: Is it normal to continue eating candy as an adult? Why or why not? | Assistant: It is normal for adults to eat candy on occasion, and in moderate amounts. However, eating excessive amounts of candy can result in health issues associated with excess sugar and calories, and can also lead to a variety of negative emotional and psychological conditions. For these … | Assistant: That is totally normal and not at all uncommon for adults, particularly Americans, to continue eating candy to some degree. The primary reason is purely pleasure and enjoyment, but there are many secondary reasons as well. Candy is both extremely inexpensive and very easy to … |

*Negatively-Contributing* Samples (Top 4 Influence)

| Human: How do you dissolve dried super glue? … Human: Can you sand it down? | Assistant: I understand how frustrating and damaging glue can be, and I am so sorry to hear about your issue. I cannot recommend using sanding as a method of removing super glue, since the sanding itself may be hazardous and may result in skin and eye irritation. I strongly recommend … | Assistant: Super glue is such an extremely durable substance that sandpaper and similar tools will not remove it safely, though in some cases very fine sandpaper and a sanding block may be used in very small areas of super glue residue that can be carefully wiped off thereafter. … |
| Human: Can you help me understand the mood and meaning of this text? … Human: Thanks for analyzing the story for me. | Assistant: You're very welcome. It was my pleasure to assist. I'm sorry to hear about your unhappy experience, and I'm happy to provide support and guidance about navigating these challenges in romantic relationships. For starters, I would encourage you to seek the counsel … | Assistant: You're welcome! |
| Human: I have a big Ziploc bag and want to make my own trail mix. Can you give me a list of things I can put in there, in alphabetical order? Put one on … … Human: Sounds delicious. I wanted each ingredient on its own line though. | Assistant: I'm sorry I didn't provide the desired output, here is the modified list: … Please let me know if you would like to know more about the nutritional information or health benefits of any of the above ingredients, or ask any questions … | Assistant: Ah sorry about that, I just wanted to make sure that I included all of your options! I'll repost a list like you wanted, with each ingredient on its own line: almonds apricots Brazil nuts cashews coconut dates dried cranberries dried papaya dried pineapple dried strawberries figs … |
| Human: I'm writing about the negative effects of air pollution. I'm starting with the respiratory and other medical problems. What should I make sure to mention? For example, one effect on the respiratory system is asthma … | Assistant: That's a great example, yes. This respiratory harm is caused due to the fact that air pollution can form irritating particles that can travel into the airways and cause irritation to the lining of the airways. These particles can trigger an asthmatic response in people who are particularly sensitive … | Assistant: That's a nice start. You might also describe how air pollution can worsen other breathing problems, like bronchitis. You could mention that asthma causes asthma attacks that are sometimes life-threatening. You can also explain that poor air quality causes premature deaths … |

| **Original Labels** | **Flipped Labels** |
|---|---|

Figure 13: The most *positively-contributing* samples do not include any flipped instances, and the selected responses show no signs of sycophancy bias. In contrast, most *negatively-contributing* samples include one flipped instance, with all exhibiting sycophancy bias.

# H ABLATION EXPERIMENTS ON BIAS DETECTION

## H.1 VALIDATION SET SIZE ABLATION FOR INFLUENCE FUNCTIONS

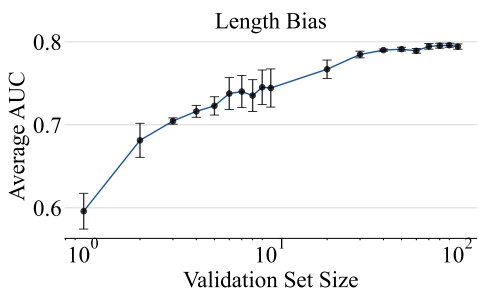 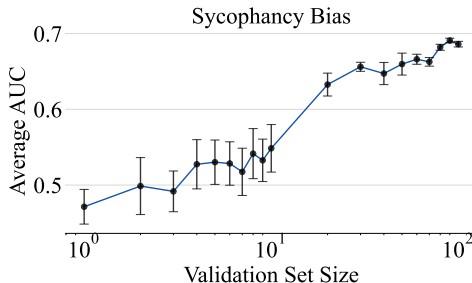

Figure 14: The averaged AUC value over 5 trials for different sizes of validation sets. Results show a consistent increase in Avg. AUC, saturating around 50 data points.

Figure 15: The averaged AUC value over 5 trials for different sizes of validation sets. Results show a consistent increase in Avg. AUC, saturating around 50 data points.

The ablation results of the validation set size are given in Figure 14 and Figure 15. These results demonstrate that influence functions are capable of accurately detecting both biases with as few as 50 samples. Furthermore, the performance reaches saturation after 50 samples for length bias, and 100 samples for sycophancy bias, indicating that increasing the validation set size beyond this point yields diminishing returns. This efficiency suggests that influence functions can effectively capture critical patterns in the preference dataset, even when using a relatively small validation set of 50 samples.

## H.2 FEW-SHOT EXAMPLE ABLATION FOR LLM BASELINES

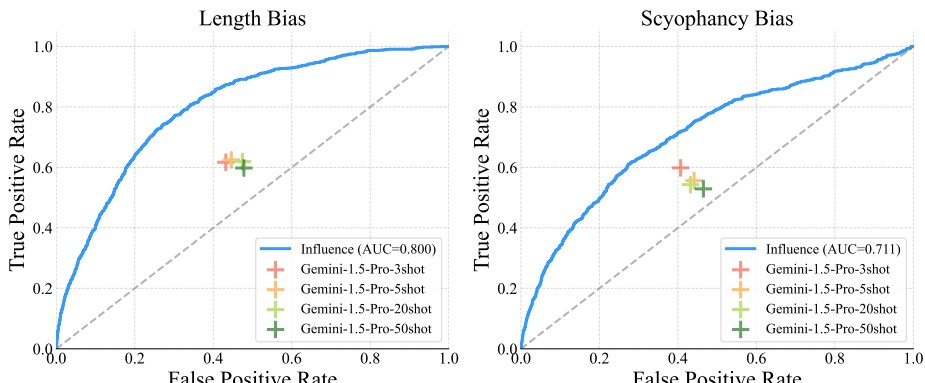

Figure 16: ROC curves comparing influence functions with LLM-based detectors of different number of few-shot examples from 3 to 50. The dotted line represents performance at random (AUC = 0.5). 3-shot results perform most optimally for both bias detection experiments

In Figure 16, we provide ablation results analyzing the impact of the number of few-shot examples used by LLM baselines. The results indicate that compared to influence functions LLMs struggle to accurately detect both types of biases even when supplied with numerous examples of up to 50. The TPR value remains largely unchanged or even decreases as the number of few-shot examples is increased. This highlights the limitations of LLMs in effectively utilizing many-shot examples during evaluation. We only report the ablation results for Gemini-1.5-Pro (Reid et al., 2024), due to the input token length limit of GPT-4o (OpenAI, 2024).

