# OpenReview forum: "Understanding Impact of Human Feedback via Influence Functions"
_ICLR.cc/2025/Conference — Submitted to ICLR 2025_

### Official Review · Reviewer_facp · 2024-10-29

**Soundness:** 3
**Presentation:** 3
**Contribution:** 2
**Rating:** 5
**Confidence:** 4

**Summary:**

This paper addresses the noise, inconsistency, and bias in the reward model of RLHF tasks. The authors introduce a method using influence functions to quantify the impact of human feedback on reward models, enhancing understanding of both feedback and the models themselves.

**Strengths:**

1.	The authors present a novel perspective on evaluating how human feedback affects the performance of reward models.

2.	They employ vector compression techniques to approximate influence functions, reducing computational costs.

3.	The paper highlights two potential applications of influence functions: detecting bias and assisting labelers.

**Weaknesses:**

1.	Evaluation Set Requirement: The method requires a high-quality evaluation set for each alignment task, ideally annotated by experts. While this constraint is noted in the "Limitations" section, it presents a significant obstacle for real-world applications.

2.	Computational Complexity: The influence function requires calculating the inverse of the Hessian matrix. Although the authors approximate it using DataInf (Eq. 6) and vector compression technology, it also necessitates summing the data in the training set, which may compromise computational precision.

3.	Bias Detection Methodology: The approach of identifying bias when the influence score exceeds a specified threshold is debatable. This threshold is difficult to determine, and not all samples surpassing it should be considered biased.
4.	Experimentation Concerns (Sec 5.2, Fig 6):

    a.	Finding expert labelers for every aspect is challenging in practical scenarios.

    b.	This method may reduce the diversity of the generated responses, potentially negatively impacting model training.

    c.	The experiment relies on training an SVM, which incurs additional computational costs and may lack accuracy.

    d.	It might be more reasonable for the weight of non-experts to be adjusted based on the attribute and value of different rewards rather than introducing another labeler.

**Questions:**

1.	Evaluation Set Size: How should the evaluation set size be determined? Is there any analysis on how performance improves with an increased number of samples?

2.	Bias Types: Have you tested your method on biases other than length and sycophancy?

3.	Metric Choice: Why does the paper report ROC curves instead of other metrics?

4.	Dependent Rewards: In real-world applications, rewards are often not independent. How does the proposed method address this in Sec 5.2?

---

> ### Author Response · Authors · 2024-11-21
> **Response to Reviewer facp [1/2]**
>
> Dear Reviewer facp, we sincerely appreciate your valuable and helpful feedback. We address each comment in detail, one by one below.
>
> ---
>
> **[W1] Evaluation Set Requirement**
>
> Thank you for your insightful comment. We acknowledge that requiring targeted evaluation sets can be a barrier to using our method (as mentioned in Remark 4.1 of the original draft). However, it is important to highlight that influence functions can still be effective even with less curated sets. To demonstrate this, we conducted additional experiments on bias detection using the original validation set of the Anthropic HH dataset, denoted as ‘Full’. This dataset was used in its unmodified, uncurated form as provided by the Anthropic HH validation set. The table below shows that influence functions effectively detect biased samples using this uncurated set compared to the high-quality, carefully filtered validation set used in our study (see Section 5.1.1 for more details).
>
> |                 | High-quality |  Full |
> |:---------------:|:------------:|:-----:|
> |   Length Bias   |     0.800    | 0.770 |
> | Sycophancy Bias |     0.711    | 0.585 |
>
> These results confirm that influence functions are robust even without well-curated validation sets, demonstrating their utility with more generalized samples. We also note that influence functions demonstrate reasonable performance even with relatively small validation sets (approximately 50-100 samples), as shown in Figure 8 and Appendix G. This suggests the potential for practical applications, using small-scale expert data.
>
> We have revised Figure 4 (with Section 5.1.2) of our draft to reflect these findings, with the changes highlighted in red.
>
> ---
>
> **[W2] Computational Complexity**
>
> We agree that computational complexity arising from computing the inverse Hessian matrix poses a fundamental challenge in utilizing influence functions. This complexity can become a major bottleneck when applying influence functions to large models and extensive datasets. However, our work demonstrates that it is feasible to leverage efficient estimation methods for influence functions to a setup involving large human feedback datasets and models with billions of parameters.
> Furthermore, we anticipate continued algorithmic advancements in this area, which, under certain assumptions about datasets and models, may enable the development of more efficient approximation methods. Our approach is compatible with any efficient computation techniques related to influence functions.
>
> ---
>
> **[W3] Bias Detection Methodology**
>
> We acknowledge that relying on a specific threshold presents limitations in threshold-based detection systems. However, this approach has been commonly utilized in prior works [1-3] to measure the quality of score metrics. To complement these limitations, we also measure threshold-independent metrics, like the AUC, to assess overall performance. Finally, we remark that systematic methods such as maximizing the F1-score or Youden’s J-statistic [4] can be employed to set more objective thresholds.
>
> ---
>
> **[W4] Experimental concerns of labeler strategy oversight experiment**
>
> Thank you for your detailed comments. We address each point below:
>
> **(a) Expert labelers for every aspect**: Our method does not require expert labelers for each fine-grained objective. The experiments in Section 5.2 serve as a proof-of-concept to demonstrate how influence functions can guide labeler strategies. In practical settings, labelers typically assess multiple aspects internally and integrate these assessments to determine their response preferences.
>
> **(b) Diversity of responses**: Our approach involves reassigning preference labels within the existing dataset without altering the responses themselves. Therefore, we believe that our method does not reduce the diversity of the generated responses. It maintains the original variety of responses while ensuring the labels are accurately applied.
>
> **(c) Computational costs and SVM accuracy**: Training the SVM for our experiment is efficient: classifying 4-dimensional vectors across 8,218 data points in mere seconds. The computational demands are minimal, and the accuracy is sufficient for our proof-of-concept.
>
> **(d) Updating labeler strategies**: Our current setup requires an expert labeler to provide ground-truth preference labels on the validation set, as our objective is to align non-expert labels with those of the experts. We welcome further clarification on the experimental setup or alternative labeling strategy update methods you have in mind. We are eager to explore these suggestions in detail and conduct additional experiments if necessary.

---

> ### Author Response · Authors · 2024-11-21
> **Response to Reviewer facp [2/2]**
>
> **[Q1] Evaluation set size**
>
> We conducted ablation experiments with various evaluation set sizes, detailed in Appendix H and illustrated in Figure 8 of the original draft. The results of these experiments indicate that influence functions perform well even with as few as 50 samples. These empirical findings suggest that efficient, curated validation sets can be achieved with approximately 50-100 samples. Increasing the size of the validation set beyond 100 results in marginal performance improvements, which tend to saturate around 100 samples.
>
> ---
>
> **[Q2] Considering different bias types than length and sycophancy**
>
> Thank you for this important question. We considered various biases identified in previous research on preference datasets. We chose to focus on length and sycophancy because they are prevalent across diverse datasets and models, including LLMs and reward models.
>
> For instance, Singhal et al. [5] noted that LLMs fine-tuned with RLHF tend to produce lengthier responses, a tendency attributable to datasets that favor longer text. Similarly, sycophancy bias is widely recognized in state-of-the-art LLMs, such as GPT-4, where datasets biased towards sycophantic responses have been shown to influence model outputs after RLHF [6]. Given these findings, we believe addressing these biases is crucial for improving dataset integrity and quality of current LLMs.
>
> Additionally, investigating other potential biases, such as self-preference bias among LLM evaluators [7], would be an interesting future direction. This bias involves evaluators favoring responses generated by themselves or similar LLM models over those produced by humans.
>
> ---
>
> **[Q3] Metric Choice**
>
> Thank you for your question. We have expanded our analysis to include additional metrics: the area under the precision-recall curve (AP) and the TNR value at TPR=0.8 (TNR80). The table below demonstrates that Influence functions consistently outperform all baselines across these metrics for both length and sycophancy bias detection. This comprehensive analysis confirms that our improvements are not at the expense of overfitting specific biases, thereby ensuring the overall robustness of our method. We have included these results, along with precision-recall curves, in Appendix E of the revised draft.
>
> |             | Length |       |       | Syco. |       |       |
> |:-----------:|:------:|:-----:|:-----:|:-----:|:-----:|:-----:|
> |             |   AUC  |   AP  | TNR80 |  AUC  |   AP  | TNR80 |
> |  Influence  |  0.800 | 0.237 | 0.664 | 0.711 | 0.121 | 0.489 |
> |  Confidence |  0.616 | 0.094 | 0.361 | 0.585 | 0.064 | 0.297 |
> |   Entropy   |  0.589 | 0.079 | 0.333 | 0.533 | 0.045 | 0.278 |
> | Mahalanobis |  0.576 | 0.082 | 0.277 | 0.560 | 0.052 | 0.237 |
> |     KNN     |  0.582 | 0.083 | 0.303 | 0.533 | 0.047 | 0.230 |
>
> ---
>
> **[Q4] Dependent Rewards**
>
> In Section 5.2, we treated the fine-grained objectives as independent to align with the framework used in the HelpSteer2 dataset [8], where different aspects of response quality are evaluated separately. We adopted HelpSteer2's methodology of modeling response quality as a linear combination of these objectives. Extending our setup to accommodate dependent objectives is an interesting direction for future research.
>
> ---
>
> [1] Kimin Lee, Kibok Lee, Honglak Lee, and Jinwoo Shin. A simple unified framework for detecting out-of-distribution samples and adversarial attacks. Advances in neural information processing systems, 31, 2018.
>
> [2] Yiyou Sun, Yifei Ming, Xiaojin Zhu, and Yixuan Li. Out-of-distribution detection with deep nearest
> neighbors. In International Conference on Machine Learning, 2022.
>
> [3] Johnson Kuan and Jonas Mueller. Model-agnostic label quality scoring to detect real-world label
> errors. International Conference on Machine Learning DataPerf Workshop, 2022.
>
> [4] Youden William John. Index for rating diagnostic tests. Cancer, 3: 32–35. 1950.
>
> [5] Singhal, Prasann, et al. A long way to go: Investigating length correlations in rlhf. arXiv preprint arXiv:2310.03716, 2023.
>
> [6] Sharma, Mrinank, et al. Towards understanding sycophancy in language models. arXiv preprint arXiv:2310.13548, 2023.
>
> [7] Wataoka, Koki, Tsubasa Takahashi, and Ryokan Ri. Self-Preference Bias in LLM-as-a-Judge. arXiv preprint arXiv:2410.21819, 2024.
>
> [8] Zhilin Wang, Yi Dong, Olivier Delalleau, Jiaqi Zeng, Gerald Shen, Daniel Egert, Jimmy J Zhang, Makesh Narsimhan Sreedhar, and Oleksii Kuchaiev. Helpsteer2: Open-source dataset for training top-performing reward models. arXiv preprint arXiv:2406.08673, 2024.

---

> > ### Comment · Reviewer_facp · 2024-11-25
> >
> > Thank you for your response and revisions. Some of my concerns have been addressed, which I appreciate.
> >
> > However, certain limitations still exist and are not easily resolvable, such as W2, W3, and W4(a).
> >
> > Therefore, I will maintain my current score.

---

> > > ### Author Response · Authors · 2024-11-26
> > >
> > > Dear Reviewer facp, thank you for your response.
> > >
> > > We are glad that some of your concerns have been addressed. However, we gently ask for further clarification for each of your remaining concerns, as it would greatly help us improve our work. Below, we summarize our responses for each remaining weakness:
> > >
> > > **[W2] Computational complexity**: We emphasize that our method significantly enhances the scalability of influence functions, achieving a 2.5x speed compared to DataInf. Specifically, we estimate influence functions on large preference datasets (up to 100k), and models with billions of parameters (e.g., Llama-3-8B) in 11 hours using a single NVIDIA A6000 GPU. Furthermore, we empirically verified our estimation method is sufficient for various applications such as bias detection and labeler strategy oversight.
> > >
> > > **[W3] Bias detection methodology**: We emphasize that influence function performances are measured through **threshold-independent** metrics such as AUC and average precision (AP) and do not rely on a fixed threshold. Also, the F1-score or Youden's J-statistic can be used to determine specific thresholds if necessary.
> > >
> > > **[W4]-(a) Expert labelers for every aspect**: To clarify, our approach does not require multiple labelers for each fine-grained sub-objective. The sub-objective scores of the HelpSteer2 dataset were solely used to **simulate preference labels**, and labeling sub-objective scores is not specifically required for utilizing influence functions to guide labelers. We kindly ask for clarification about the remaining weakness you have in mind.
> > >
> > > Thank you for your time and effort in reviewing our work. We remain committed to addressing any remaining concerns and would greatly appreciate your guidance on any limitations or issues that require further clarification or improvement. Thank you again for your valuable feedback.
> > >
> > > Best Regards,
> > > Authors

---

> ### Author Response · Authors · 2024-11-25
> **Gentle Reminder**
>
> Dear Reviewer facp,
>
> Thank you again for your time and efforts in reviewing our paper.
>
> As the discussion period draws close, we kindly remind you that two days remain for further comments or questions. We would appreciate the opportunity to address any additional concerns you may have before the discussion phase ends.
>
> Thank you very much!
>
> Many thanks,
> Authors

---

### Official Review · Reviewer_dPes · 2024-10-31

**Soundness:** 4
**Presentation:** 4
**Contribution:** 2
**Rating:** 8
**Confidence:** 2

**Summary:**

The paper describes the use of influence functions to evaluate the influence of preference samples in reward modeling. The authors apply recent methods for computing influence functions for large neural networks/LLM to reward model training. By using a validation set demonstrating preferences for specific qualities, the influence of examples is highlighted. The authors also propose a compression scheme to enable efficient storage of gradient vectors for dot products. The authors apply the method for the detection of biased data samples, and outline guidance of human labelers.

**Strengths:**

Positive:
- The paper is well written and polished, I found it easy to follow
- The main presented methods seems to work well, and is evaluated appropriately
- The different use cases (bias detection and user guidance) are presented well and relevant
- Limitations are discussed (such as the dependence on validation set composition), and runs (although limited) ablations on the effects of different compositions
- The quality of experiments seems high, both qualitative and quantitative results are generally convincing
- Description of experiments/data sets is sufficient for reproducibility
- The paper presents a nice application of influence functions

Overall, I enjoyed reading the paper, and am convinced by its research quality. While strict technical novelty or theoretical insights are limited, I think it still provides a meaningful contribution to the field.

**Weaknesses:**

Negative:
- The methdological novelty is limited, i.e. just the application of existing methods to a new use case
- The dependency of the method on validation set  composition seems like a core limitation, which might make wide application difficult
- The use case of user guidance is just on a proof-of concept level, and it’s unclear how it would transfer to an actual human scenario, however (again) this is appropriately acknowledged by the authors

**Questions:**

Questions/Needs clarification:
- You mention “L.140 been extended beyond the model parameter to any univariate quantity of interest f(θ), such as validation loss”. Can you make the reference to using a validation dataset more explicit? Instead of the chained references at the section below (Koh&Liang, …)
- How did you decide on the Mahalanobis distance as a baseline measure? Has this been considered before for this kind of task?

---

> ### Author Response · Authors · 2024-11-21
> **Response to Reviewer dPes [1/2]**
>
> Dear Reviewer dPes, we sincerely appreciate your valuable and helpful feedback. We address each comment in detail, one by one below.
>
> ---
>
> **[W1] Novelty of our work**
>
> First, thank you for noting that this work provides a meaningful contribution to the field. Regarding the novelty of our method, we would like to further elaborate on our contributions.
>
> **Efficient estimation method**: The influence function computation method suggested by Koh and Liang [1] is inefficient for application to modern LLMs or large preference datasets. We address this computational challenge by implementing the vector compression method OPORP (One Permutation + One Random Projection) [2] alongside DataInf [3], significantly enhancing computational efficiency.
>
> **Practical value of approach**: Our work demonstrates practical applications using influence functions: enhancing the interpretability of human feedback. These applications address the growing need to provide reliable oversight to modern LLMs. The significance of our research topic was additionally acknowledged by reviewers CzsY, Wn7U, and roz9.
>
> **Bias detection**: We have applied influence functions to detect prevalent types of labeler bias (i.e., length and sycophancy biases) in RLHF. Our approach demonstrates superior performance in identifying biased samples compared to other baselines, including GPT-4o.
>
> **Labeler strategy oversight**: Through a proof-of-concept experiment, we provide preliminary evidence that influence functions can improve non-expert labeling strategies by leveraging small sets of expert-labeled samples. This suggests a scalable path for enhancing oversight using influence functions.
>
> We hope this clarifies the novelty and practical value of our work. Thank you for the opportunity to address this point.
>
> ---
>
> **[W2] Dependency on validation set composition**
>
> Thank you for your insightful comment. We acknowledge that requiring targeted evaluation sets can be a barrier to using our method (as mentioned in Remark 4.1 of the original draft). However, it is important to highlight that influence functions can still be effective even with less curated sets. To demonstrate this, we conducted additional experiments on bias detection using the original validation set of the Anthropic HH dataset, denoted as ‘Full’. This dataset was used in its unmodified, uncurated form as provided by the Anthropic HH validation set. The table below shows that influence functions effectively detect biased samples using this uncurated set compared to the high-quality, carefully filtered validation set used in our study (see Section 5.1.1 for more details).
>
> |                 | High-quality |  Full |
> |:---------------:|:------------:|:-----:|
> |   Length Bias   |     0.800    | 0.770 |
> | Sycophancy Bias |     0.711    | 0.585 |
>
> These results confirm that influence functions are robust even without well-curated validation sets, demonstrating their utility with more generalized samples. We also note that influence functions demonstrate reasonable performance even with relatively small validation sets (approximately 50-100 samples), as shown in Figure 8 and Appendix G. This suggests the potential for practical applications, using small-scale expert data.
>
> We have revised Figure 4 (with Section 5.1.2) of our draft to reflect these findings, with the changes highlighted in red.
>
> ---
>
> **[W3] labeler guidance experiment remains as proof-of-concept**
>
> Thank you for your insightful feedback. Our labeling oversight experiment serves as a proof-of-concept demonstrating how influence functions might guide labelers. As noted in the 'Limitations' section, we acknowledge the limitation that our setup does not perfectly model real-world conditions. However, despite these constraints, we believe our results provide valuable preliminary evidence that influence functions can be beneficial. We hope our work encourages further research to test influence functions in more realistic settings. We have also revised the introduction section of our draft for clarification.

---

> ### Author Response · Authors · 2024-11-21
> **Response to Reviewer dPes [2/2]**
>
> **[Q1] Clarifying reference on L.140**
>
> Thank you for your detailed comment. We have revised our draft accordingly.
>
> ---
>
> **[Q2] Reason for Mahalabobis distance baseline choice**
>
> The Mahalanobis distance is a proven metric for detecting out-of-distribution (OOD) samples [4] and noisy labels [5]. Given that the labeler bias detection problem shares similarities with these issues, we utilized it as a baseline measure.
>
> ---
>
> [1] Pang Wei Koh and Percy Liang. Understanding black-box predictions via influence functions. In International conference on machine learning, 2017.
>
> [2] Ping Li and Xiaoyun Li. Oporp: One permutation+ one random projection. In ACM SIGKDD Conference on Knowledge Discovery and Data Mining, 2023.
>
> [3] Yongchan Kwon, Eric Wu, Kevin Wu, and James Zou. Datainf: Efficiently estimating data influence
> in lora-tuned llms and diffusion models. In International Conference on Learning Representa-
> tions, 2024.
>
> [4] Kimin Lee, Kibok Lee, Honglak Lee, and Jinwoo Shin. A simple unified framework for detecting out-of-distribution samples and adversarial attacks. Advances in neural information processing systems, 31, 2018.
>
> [5] Lee, Kimin, et al. Robust inference via generative classifiers for handling noisy labels. International conference on machine learning. PMLR, 2019.

---

> > ### Comment · Reviewer_dPes · 2024-11-21
> >
> > I thank the authors for their comments and revision, in particular for the additional experiments.
> > I was generally convinced by the work before the revision, and this has not changed significantly.
> > In accordance, i will probably keep my score.

---

> > > ### Author Response · Authors · 2024-11-22
> > >
> > > Dear Reviewer dPes, Thank you for your response and for your thoughtful evaluation of our work. We greatly appreciate your positive feedback and the time you have dedicated to reviewing our paper.
> > >
> > > If there are any remaining concerns or suggestions for improvement, we would be more than happy to address them.
> > >
> > > Thank you once again for your support and valuable feedback.

---

### Official Review · Reviewer_Wn7U · 2024-11-03

**Soundness:** 3
**Presentation:** 3
**Contribution:** 3
**Rating:** 6
**Confidence:** 2

**Summary:**

This paper discusses using influence funcitons to identify potential bias of human labelers in human feedback data for LLM alignment. To enable effcient estimation of incluence functions under the LLM context, DataInf and OPORP are introduced to ease the burden of computation and storage respectively. Empirical results demonstrate that the proposed method can effectively identify lengthy and sycophancy bias compared to baselines and querying powerful LLMs. Specially, the authors show that influence functions can help align non-expert labelers' strategies to experts' labeling strategy, which I believe is important for preference data annotation in LLM alignment.

**Strengths:**

1. The discussed topic is practical and important.

2. Through approximation and compression, the proposed method can estimate influence functions efficiently.

3. Empirical results are promising (although no comparison to concurrent works)

**Weaknesses:**

1. The proposed method mainly integrates existing techniques to the LLM context (although I do not think this is a serious issue as long as they work properly)

2. Mistakes in fig.4 legend (AUC=0.8 for both Concise and Verbose?)

3. Empirical study only show the efficacy in identifying bias, but discussion on data quality improvement with such detechtion is missing.

**Questions:**

1. "we reduce the size of a single gradient vector from 160MB (42M dimensions) to 256KB (65K dimensions)"

How do you calculate the size of gradient vectors? Why is the gradient vector size 160MB (Llama-3-8B in your experiment has around 15GB model size)

2. Could you provide some performance improvement results on real-world dataset after filtering biased data with incluence functions?

3. I notice you only use four dimensions of Helpsteer2 in section 5.2.1 (the four-dimensional weight vector]). Why do you make such alteration?

---

> ### Author Response · Authors · 2024-11-21
> **Response to Reviewer Wn7U [1/2]**
>
> Dear Reviewer Wn7U, we sincerely appreciate your valuable and helpful feedback. We have carefully considered your comments, and provide responses below.
>
> ---
>
> **[S3] Promising empirical results**
>
> Thank you for acknowledging the strength of our empirical results. If you have any specific concurrent works in mind, we would be happy to conduct additional experiments for comparison. Additionally, in our revised draft, we have included comparisons with another baseline, KNN [1], where our influence functions significantly outperform it.
>
> ---
>
> **[W1] Novelty of our work**
>
> Thank you for also kindly noting that the technical novelty is not a serious issue. We would like to elaborate on our contributions further.
>
> **Efficient estimation method**: The influence function computation method suggested by Koh and Liang [2] is inefficient for application to modern LLMs or large preference datasets. We address this computational challenge by implementing the vector compression method OPORP (One Permutation + One Random Projection) [3] alongside DataInf [4], significantly enhancing computational efficiency.
>
> **Practical value of approach**: Our work demonstrates practical applications using influence functions: enhancing the interpretability of human feedback. These applications address the growing need to provide reliable oversight to modern LLMs. The significance of our research topic was additionally acknowledged by reviewers CzsY, roz9.
>
> **Bias detection**: We have applied influence functions to detect prevalent types of labeler bias (i.e., length and sycophancy biases) in RLHF. Our approach demonstrates superior performance in identifying biased samples compared to other baselines, including GPT-4o.
>
> **Labeler strategy oversight**: Through a proof-of-concept experiment, we provide preliminary evidence that influence functions can improve non-expert labeling strategies by leveraging small sets of expert-labeled samples. This suggests a scalable path for enhancing oversight using influence functions.
>
> We hope this clarifies the novelty and practical value of our contributions. Thank you for the opportunity to address this point.
>
> ---
>
> **[W2] Mistakes in Fig 4 legend**
>
> Thank you for catching this mistake. We have corrected Fig. 4 accordingly. The AUC values are 0.800 for the *Concise* set and 0.202 for the *Verbose* set.
>
> ---
>
> **[W3 & Q2] Discussion of Data Quality Improvement on real-world datasets**
>
> Thank you very much for your constructive feedback. To assess whether our bias detection method enhances data quality and reward model performance in real-world datasets, we conducted additional experiments outlined in Section 5.1. We utilized a 15k helpful subset of the Anthropic HH dataset without flipping any labels to introduce synthetic bias. Using our curated Concise validation set, designed to capture length bias, we calculated the influence for training samples and identified the top 100 samples exhibiting positive influence. Upon manual inspection (by authors) to verify label accuracy, we found a significant discrepancy in labeling compared to a control group of 100 randomly selected samples:
>
> |         | Mislabeled | Correct | Tie |
> |:-------:|:----------:|:-------:|:---:|
> | Top 100 |     47     |    38   |  15 |
> |  Random |     13     |    69   |  18 |
>
> This result indicates that the top-influenced samples, predominantly longer responses, were more frequently mislabeled. To test the impact on model performance, we trained reward models on a dataset corrected for the 47 mislabeled samples. The reward model trained on this refined dataset achieved a score of 56.1 on Rewardbench [5], showing a modest improvement over the 54.0 scored by models trained on the unaltered dataset. We believe that these results highlight the potential effectiveness of our method when applied to real-world datasets.
>
> We will open-source the human-labeled results, including both the original and updated training datasets.
>
> ---

---

> ### Author Response · Authors · 2024-11-21
> **Response to Reviewer Wn7U [2/2]**
>
> **[Q1] Reason why gradient vector dimension is 42M**
>
> The size of a single gradient vector is 42M dimensions (160MB for storage) because we utilized Low-Rank Adaptation [6] fine-tuning with a rank of 16 as mentioned in the ‘Reward model training’ paragraph of Section 5.1.1. For clarity, we have included a footnote explaining this in Section 4.2 of our revised draft.
>
> ---
>
> **[Q3] Reason for using 4 dimensions of HelpSteer2**
>
> Thank you for this detailed question. As you have correctly noted, HelpSteer2 [7] evaluates responses based on five criteria: helpfulness, correctness, coherence, complexity, and verbosity. In our study, we utilize four of these criteria, excluding ‘helpfulness.’ This exclusion is because the helpfulness score is considered an overall evaluation of the response, unlike the other four criteria that measure specific sub-aspects (refer to Appendix G.3: Per-axis ratings in the HelpSteer2 paper). We believe the helpfulness score is unnecessary for our experimental motivation, which requires labelers to make decisions based on more fine-grained objectives. This explanation has been added to Appendix B.2 in red for further clarification.
>
> ---
>
> [1] Yiyou Sun, Yifei Ming, Xiaojin Zhu, and Yixuan Li. Out-of-distribution detection with deep nearest
> neighbors. In International Conference on Machine Learning, 2022.
>
> [2] Pang Wei Koh and Percy Liang. Understanding black-box predictions via influence functions. In International conference on machine learning, 2017.
>
> [3] Ping Li and Xiaoyun Li. Oporp: One permutation+ one random projection. In ACM SIGKDD Conference on Knowledge Discovery and Data Mining, 2023.
>
> [4] Yongchan Kwon, Eric Wu, Kevin Wu, and James Zou. Datainf: Efficiently estimating data influence
> in lora-tuned llms and diffusion models. In International Conference on Learning Representa-
> tions, 2024.
>
> [5] Nathan Lambert, Valentina Pyatkin, Jacob Morrison, LJ Miranda, Bill Yuchen Lin, Khyathi Chandu,
> Nouha Dziri, Sachin Kumar, Tom Zick, Yejin Choi, et al. Rewardbench: Evaluating reward
> models for language modeling. arXiv preprint arXiv:2403.13787, 2024.
>
> [6] Edward J Hu, Phillip Wallis, Zeyuan Allen-Zhu, Yuanzhi Li, Shean Wang, Lu Wang, Weizhu Chen, et al. Lora: Low-rank adaptation of large language models. In International Conference on Learning Representations, 2022.
>
> [7] Zhilin Wang, Yi Dong, Olivier Delalleau, Jiaqi Zeng, Gerald Shen, Daniel Egert, Jimmy J Zhang, Makesh Narsimhan Sreedhar, and Oleksii Kuchaiev. Helpsteer2: Open-source dataset for training top-performing reward models. arXiv preprint arXiv:2406.08673, 2024.

---

> ### Author Response · Authors · 2024-11-25
> **Gentle Reminder**
>
> Dear Reviewer Wn7U,
>
> Thank you again for your time and efforts in reviewing our paper.
>
> As the discussion period draws close, we kindly remind you that two days remain for further comments or questions. We would appreciate the opportunity to address any additional concerns you may have before the discussion phase ends.
>
> Thank you very much!
>
> Many thanks,
> Authors

---

> > ### Comment · Reviewer_Wn7U · 2024-11-25
> >
> > Thanks for your response, especially the experiment on the refined dataset. Now my concerns have been addressed.

---

> > > ### Author Response · Authors · 2024-11-25
> > > **Thank you for your response**
> > >
> > > Dear Reviewer Wn7U, Thank you for your response. We are happy to hear that your concerns have been addressed.
> > >
> > > If there are any remaining concerns or suggestions for improvement, we would be more than happy to address them.
> > >
> > > Thank you once again for your valuable feedback.

---

### Official Review · Reviewer_71Vm · 2024-11-03

**Soundness:** 3
**Presentation:** 3
**Contribution:** 1
**Rating:** 5
**Confidence:** 3

**Summary:**

This paper presents a novel approach to evaluate and refine the reward models in RLHF by employing influence functions. The authors argue that human feedback, which is integral to training reward models for LLMs, can be noisy and biased, leading to misaligned rewards. To mitigate this, they introduce a compute-efficient method to approximate influence functions, allowing for the measurement of individual data points' impact on reward model performance. The paper claims two main applications: detecting labeler bias in feedback datasets and guiding labelers to align more closely with expert feedback. The authors demonstrate the effectiveness of their approach through experiments and argue that it enhances feedback interpretability and contributes to scalable oversight in RLHF.

**Strengths:**

The paper introduces a novel approach to enhance the interpretability of reward models. By applying influence functions, the authors provide a method to quantify the impact of individual feedback on the model's performance, offering insights into how human feedback shapes the reward model's outcomes.

The idea of using influence functions to measure the impact of human feedback is innovative and has the potential to contribute to the broader goal of scalable oversight in RLHF. This approach can help in detecting and mitigating labeler bias, which is a common challenge in training robust and aligned AI systems.

**Weaknesses:**

As far as I am concerned, the authors simpy apply the approach in [1] to the reward modeling scenario, which greatly limits the novelty of the paper. I suggest that the author summarize the main contributions.

While the experiments show promise, establishing reward models with various LLMs and evaluating with more downstream alignment tasks, such as direct alignment algorithms, could further validate the generalizability of the approach.

Reference:
[1] Koh P W, Liang P. Understanding black-box predictions via influence functions[C]//International conference on machine learning. PMLR, 2017: 1885-1894.

**Questions:**

Could you provide further insights into how the method performs when handling intentionally misleading or adversarial feedback? Are there specific patterns or checks in place to identify and mitigate such cases?

Please provide more specifics on how the method scales with increasingly large datasets. What computational trade-offs may arise as the dataset grows?

---

> ### Author Response · Authors · 2024-11-21
> **Response to Reviewer 71Vm [1/2]**
>
> Dear Reviewer 71Vm, we sincerely appreciate your valuable and helpful feedback. We address each comment in detail, one by one below.
>
> ---
>
> **[W1] Novelty of our work**
>
> We would like to clarify and emphasize our contributions as follows:
>
> **Efficient estimation method**: The influence function computation method suggested by Koh and Liang [1] is inefficient for application to modern LLMs or large preference datasets. We address this computational challenge by implementing the vector compression method OPORP (One Permutation + One Random Projection) [2] alongside DataInf [3], significantly enhancing computational efficiency.
>
> **Practical value of approach**: Our work demonstrates practical applications using influence functions: enhancing the interpretability of human feedback. These applications address the growing need to provide reliable oversight to modern LLMs. The significance of our research topic was additionally acknowledged by reviewers CzsY, Wn7U, and roz9.
>
> **Bias detection**: We have applied influence functions to detect prevalent types of labeler bias (i.e., length and sycophancy biases) in RLHF. Our approach demonstrates superior performance in identifying biased samples compared to other baselines, including GPT-4o.
>
> **Labeler strategy oversight**: Through a proof-of-concept experiment, we provide preliminary evidence that influence functions can improve non-expert labeling strategies by leveraging small sets of expert-labeled samples. This suggests a scalable path for enhancing oversight using influence functions.
>
> We hope this clarifies the novelty and practical value of our contributions. Thank you for the opportunity to address this point.
>
> ---
>
> **[W2] Evaluating influence functions for different models**
>
> Thank you for your valuable feedback. To assess the robustness of the influence functions across different models, we added a new model, Gemma-2-9B [4], in the length bias detection experiment. As shown in the table below, the influence functions demonstrate competitive performance for both Llama and Gemma, outperforming the GPT-40 baseline. We report the TPR values at tied FPR rates with GPT-4o (FPR=0.283)
>
> |  Method  |  TPR  |
> |:-------------:|:-----:|
> |  GPT-4o  | 0.696 |
>  |  **Influence functions (Llama-3-8B)**  | 0.749 |
> |  **Influence functions (Gemma-2-9B)**  | 0.720 |
>
> We also recognize the importance of evaluating more downstream alignment methods, such as DPO [5]. Although time constraints prevented us from testing our method with these algorithms, we plan to explore this in future studies to provide a more comprehensive validation of our approach.
>
> ---
>
> **[Q1] Handling misleading feedback**
>
> Thank you for the intriguing question. In our labeler bias detection experiments, we specifically aim to identify intentionally misleading feedback with certain biases, such as those favoring lengthy or flattering responses. Our method has demonstrated strong detection performance, effectively identifying and mitigating such biases. We believe that our method can be effectively applied to other types of biases and adversarial feedback.

---

> ### Author Response · Authors · 2024-11-21
> **Resposne to Reviewer 71Vm [2/2]**
>
> **[Q2] Scalability of method regarding dataset size**
>
> We thank the reviewer for this important question. As shown in Figure 5 of our original draft, time consumption scales linearly with dataset sizes up to 100k samples on a single NVIDIA A6000 GPU. **This linear scaling is expected to extend to even larger datasets**, as the primary computational cost lies in the backpropagation step, which is proportional to dataset size. Importantly, this assumes adequate storage for the compressed gradient vectors of the training and validation datasets. By employing OPORP, we have compressed gradient vectors to $2^{16}$ dimensions, facilitating the storage of millions of samples, e.g., 5M samples require approximately 1TB of storage. If even larger datasets are considered, smaller compression dimensions could be used to further reduce storage needs, though at the cost of increased estimation variance, which according to the OPORP paper, is inversely proportional to the compressing dimension $\text{K}$ ($\text{Var} \propto 1/\text{K}$) [2].
>
> ---
>
> [1] Pang Wei Koh and Percy Liang. Understanding black-box predictions via influence functions. In International conference on machine learning, 2017.
>
> [2] Ping Li and Xiaoyun Li. Oporp: One permutation+ one random projection. In ACM SIGKDD Conference on Knowledge Discovery and Data Mining, 2023.
>
> [3] Yongchan Kwon, Eric Wu, Kevin Wu, and James Zou. Datainf: Efficiently estimating data influence
> in lora-tuned llms and diffusion models. In International Conference on Learning Representa-
> tions, 2024.
>
> [4] Team, Gemma, et al. Gemma 2: Improving open language models at a practical size, arXiv: 2408.00118, 2024.
>
> [5] Rafael Rafailov, Archit Sharma, Eric Mitchell, Stefano Ermon, Christopher D. Manning, Chelsea Finn. Direct preference optimization: Your language model is secretly a reward model. Advances in Neural Information Processing Systems, 2024.

---

> ### Author Response · Authors · 2024-11-25
> **Gentle Reminder**
>
> Dear Reviewer 71Vm,
>
> Thank you again for your time and efforts in reviewing our paper.
>
> As the discussion period draws close, we kindly remind you that two days remain for further comments or questions. We would appreciate the opportunity to address any additional concerns you may have before the discussion phase ends.
>
> Thank you very much!
>
> Many thanks,
> Authors

---

> > ### Comment · Reviewer_71Vm · 2024-11-25
> >
> > Thanks for your response and clarification. Part of my concerns have been addressed, however, I still think the contribution of the paper is limited, so I tend to keep my score.

---

> > > ### Author Response · Authors · 2024-11-25
> > >
> > > Dear Reviewer 71Vm, Thank you for your response and for taking the time to review our work.
> > >
> > > We would like to emphasize that our newly proposed method can effectively address an important topic, as recognized by reviewers Wn7U, CzsY, roz9, and dPes. Our work also introduces efficient estimation methods, making them highly practical and scalable to implement on large preference datasets, as specifically noted by reviewers roz9 and CzsY.
> > >
> > > We remain committed to addressing any remaining concerns and would greatly appreciate your guidance on any limitations or issues that require further clarification or improvement.
> > > Thank you again for your valuable feedback.
> > >
> > > Best regards,
> > > Authors

---

### Official Review · Reviewer_CzsY · 2024-11-04

**Soundness:** 3
**Presentation:** 3
**Contribution:** 3
**Rating:** 8
**Confidence:** 5

**Summary:**

The paper addresses the challenge of noise and misalignment in human feedback within RLHF, which can lead to flawed reward signals. By using influence functions, it quantifies the impact of individual data points on reward model performance. For each data point, the influence function captures how parameter adjustments would occur when the weight of that point changes. The approach assumes a small validation set is available to calculate overall influence and utilizes an approximate inverse Hessian to enable efficient computation. Results indicate improved AUC in addressing length bias and sycophancy bias compared to GPT-4o and Gemini models.

Additionally, the paper explores the potential of guiding non-expert labelers to align more closely with expert labeling. It assumes that labelers use a list of linearly combined, weighted sub-scores in their scoring process. By identifying data points that negatively impact alignment, non-expert labelers can adjust weights and improve their performance. Results suggest that with a few samples, non-expert labelers demonstrate notable improvement.

**Strengths:**

This paper excels in identifying data points that negatively influence reward models through the application of influence functions. Given the common issue of noise in human-labeled data, particularly in RLHF, this approach enhances transparency and is valuable in managing labels from non-expert labelers. The methodology effectively addresses computational challenges using an approximation function, which is highly practical aiming for reliable AI.

The experiments are well-defined, and the performance results are convincing. The paper is well-articulated regarding both analytical rigor and technical details. Additionally, the authors explore ways to improve labeler performance, which is a good extension of the study.

**Weaknesses:**

While the application of influence functions to assess the contribution of individual data points is compelling, the methodology relies heavily on manually curated validation sets, as evidenced by the ablation experiments. There are two primary concerns:

1) Dependence on Domain Knowledge: The construction of validation sets requires domain-specific knowledge, which could limit generalizability. Although addressing verbosity and sycophancy is valuable, the methodology appears capable of handling other issues if suitable validation sets are available. However, this adaptability is currently contingent on creating each validation set by hand.

2) Risk of Conflicting Improvements: It is unclear how the approach manages potential conflicts between multiple improvement directions. For instance, in the main results (Figure 2), Influence shows higher TPR than GPT-4o at the same FPR, which is promising. However, it raises the question: Does this improvement come at the expense of other metrics? To fully assess the model’s overall robustness, it would be beneficial to provide additional statistics comparing the base model and the Influence-based approach to ensure that the latter does not merely overfit to specific biases, such as length.

Furthermore, as noted in the limitations, the feasibility of improving labelers’ performance in real-world settings may be limited, especially when human labelers’ judgments involve complex decision-making beyond linear reward models. The authors could strengthen their argument by including results from human subjects whose labeling strategies demonstrably improve with guidance. Alternatively, if the current scope remains focused on bias correction within datasets, it would be beneficial to clarify this in the introduction to avoid misleading interpretations about broader applicability.

**Questions:**

1) Will focusing on improving a single aspect of bias impact other performance metrics?

2) Beyond runtime speedup, does approximating the inverse Hessian lead to any performance degradation?

3) Could you provide insights on why Entropy underperforms significantly in addressing Sycophancy Bias relative to other baselines?

4) In Appendix B2, five weights are selected for B. Could you elaborate on the rationale for these specific weights? Were they chosen randomly, and are any particularly extreme?

**Details Of Ethics Concerns:**

No concerns.

---

> ### Author Response · Authors · 2024-11-21
> **Response to Reviewer CzsY [1/2]**
>
> Dear Reviewer CzsY, we sincerely appreciate your valuable and helpful feedback. We address each comment in detail, one by one below.
>
> ---
>
> **[W1] Dependence on Domain Knowledge**
>
> Thank you for your insightful comment. We acknowledge that requiring targeted evaluation sets can be a barrier to using our method (as mentioned in Remark 4.1 of the original draft). However, it is important to highlight that influence functions can still be effective even with less curated sets. To demonstrate this, we conducted additional experiments on bias detection using the original validation set of the Anthropic HH dataset, denoted as ‘Full’. This dataset was used in its unmodified, uncurated form as provided by the Anthropic HH validation set. The table below shows that influence functions effectively detect biased samples using this uncurated set compared to the high-quality, carefully filtered validation set used in our study (see Section 5.1.1 for more details).
>
> |                 | High-quality |  Full |
> |:---------------:|:------------:|:-----:|
> |   Length Bias   |     0.800    | 0.770 |
> | Sycophancy Bias |     0.711    | 0.585 |
>
> These results confirm that influence functions are robust even without well-curated validation sets, demonstrating their utility with more generalized samples. We also note that influence functions demonstrate reasonable performance even with relatively small validation sets (approximately 50-100 samples), as shown in Figure 8 and Appendix G. This suggests the potential for practical applications, using small-scale expert data.
>
> We have revised Figure 4 (with Section 5.1.2) of our draft to reflect these findings, with the changes highlighted in red.
>
> ---
>
> **[W2 & Q1] Additional metrics for experiments**
>
> Thank you for your suggestions. We have expanded our analysis to include additional metrics: the area under the precision-recall curve (AP) and the TNR value at TPR=0.8 (TNR80). The table below demonstrates that Influence functions consistently outperform all baselines across these metrics for both length and sycophancy bias detection. This comprehensive analysis confirms that our improvements are not at the expense of overfitting specific biases, thereby ensuring the overall robustness of our method. We have included these results, along with precision-recall curves, in Appendix E of the revised draft.
>
> |             | Length |       |       | Syco. |       |       |
> |:-----------:|:------:|:-----:|:-----:|:-----:|:-----:|:-----:|
> |             |   AUC  |   AP  | TNR80 |  AUC  |   AP  | TNR80 |
> |  Influence  |  0.800 | 0.237 | 0.664 | 0.711 | 0.121 | 0.489 |
> |  Confidence |  0.616 | 0.094 | 0.361 | 0.585 | 0.064 | 0.297 |
> |   Entropy   |  0.589 | 0.079 | 0.333 | 0.533 | 0.045 | 0.278 |
> | Mahalanobis |  0.576 | 0.082 | 0.277 | 0.560 | 0.052 | 0.237 |
> |     KNN     |  0.582 | 0.083 | 0.303 | 0.533 | 0.047 | 0.230 |
>
> ---
>
> **[W3] Limitations of labeling strategy guidance experiment**
>
> Thank you for your insightful feedback. Our labeling oversight experiment serves as a proof-of-concept demonstrating how influence functions might guide labelers. As noted in the 'Limitations' section, we acknowledge the limitation that our setup does not perfectly model real-world conditions. However, despite these constraints, we believe our results provide valuable preliminary evidence that influence functions can be beneficial. We hope our work encourages further research to test influence functions in more realistic settings. We have also revised the introduction section of our draft for clarification.
>
> ---
>
> **[Q2] Performance degradation of hessian approximation**
>
> Thank you for this important question. The hessian approximation using DataInf [1] does come with approximation error. We could not perform an error analysis using our method as computing Hessians is computationally infeasible for large language models with sizes that we consider. Instead, we refer the reviewer to the DataInf paper (Section 4.1, Figure 1), which evaluates approximation error on a smaller scale using the RoBERTA-large model [2] (355M parameters). According to their experiments, DataInf achieves a correlation of approximately 0.35-0.65 with the exact influence estimate on the GLUE dataset [3]. Although this correlation does not reach optimality, it surpasses that of other estimation methods available to us. We therefore utilize DataInf to estimate influence functions, accepting this compromise as the most effective available solution within our computational constraints.

---

> ### Author Response · Authors · 2024-11-21
> **Response to Reviewer CzsY [2/2]**
>
> **[Q3] Underperformance of Entropy**
>
> We expect that Entropy shows poor performance in detecting sycophancy bias primarily because this metric does not use a validation set, which is crucial for identifying sycophancy bias. Additionally, unlike other baselines that measure scores based on activations from intermediate layers of the reward model, Entropy relies solely on logit values. This approach may fail to capture the more complex patterns in the data that are essential for recognizing sycophancy bias.
>
> ---
>
> **[Q4] Procedure of weight selection**
>
> The five initial weights for model B were selected through a random procedure, with specific considerations in mind. Firstly, we aimed to ensure that the initial labeling accuracy of the model fell within the 70-80% range. Additionally, we avoided weights that were too similar, to ensure that the selected weights captured distinct aspects of our fine-grained objectives. We have updated Appendix B.2 in our draft to include these details.
>
> ---
>
> [1] Yongchan Kwon, Eric Wu, Kevin Wu, and James Zou. Datainf: Efficiently estimating data influence
> in lora-tuned llms and diffusion models. In International Conference on Learning Representa-
> tions, 2024.
>
> [2] Liu, Yinhan. Roberta: A robustly optimized bert pretraining approach. arXiv preprint arXiv:1907.11692 364, 2019.
>
> [3] Alex Wang, Amanpreet Singh, Julian Michael, Felix Hill, Omer Levy, and Samuel R Bowman. Glue: A multi-task benchmark and analysis platform for natural language understanding. arXiv preprint arXiv:1804.07461, 2018.

---

> > ### Comment · Reviewer_CzsY · 2024-11-22
> >
> > Thank you for addressing my concerns and providing the updates and additional experiments. I appreciate the effort, and since my concerns have been resolved, I am happy to raise my score.

---

> > > ### Author Response · Authors · 2024-11-22
> > >
> > > Dear Reviewer CzsY,
> > > Thank you for your response. We are pleased to hear that your concerns have been resolved and sincerely appreciate your decision to raise the score.
> > > If there are any remaining concerns or suggestions for improvement, we would be more than happy to address them.

---

### Official Review · Reviewer_roz9 · 2024-11-04

**Soundness:** 3
**Presentation:** 3
**Contribution:** 3
**Rating:** 6
**Confidence:** 3

**Summary:**

The paper presents a novel approach to analyzing the effects of human feedback in RLHF. It introduces influence functions to quantify the impact of individual feedback on the performance of reward models, which can help detect biases and improve labeling strategies in RLHF systems. Two primary applications of influence functions are highlighted: detecting biases like length and sycophancy in human feedback and guiding labelers to align their feedback with expert standards.

**Strengths:**

1. The use of influence functions to analyze the impact of human feedback is a promising direction that adds a layer of interpretability in RLHF, which is essential for aligning LLMs with human values.
2. The paper introduces a compute-efficient method that enables scalable application of influence functions, potentially reducing computational demands by 2.5 times, a significant improvement over previous methods.
3. The methodology, experimental design, and results are presented clearly, with well-structured figures and examples to support the claims.

**Weaknesses:**

1. The study’s limitations in real-world scenarios, where expert and non-expert labelers may not share sub-objective scores, could reduce the generalizability of the approach.
2.  While the paper shows effectiveness in detecting length bias, sycophancy bias remains challenging, as it involves understanding nuanced human agreement tendencies that may vary by context.

**Questions:**

1. Could the reliance on targeted validation sets be reduced by adapting influence functions to work with more generalized validation samples?
2. How does the approach scale with increasingly large datasets or models beyond the experiments presented, and would there be additional trade-offs in compute efficiency?

---

> ### Author Response · Authors · 2024-11-21
> **Response to Reviewer roz9**
>
> Dear Reviewer roz9, we sincerely appreciate your valuable and helpful feedback. We address each comment in detail, one by one below.
>
> ---
>
> **[W1] Limitations in sharing sub-objective scores**
>
> Thank you for your insightful feedback. Our labeling oversight experiment serves as a proof-of-concept demonstrating how influence functions might guide labelers. As noted in the 'Limitations' section, we acknowledge the limitation that our setup does not perfectly model real-world conditions. However, despite these constraints, we believe our results provide valuable preliminary evidence that influence functions can be beneficial. We hope our work encourages further research to test influence functions in more realistic settings. We have revised the introduction section of our draft to make it clear that our labeler strategy guidance experiment is a proof-of-concept.
>
> ---
>
> **[W2] Sycophancy bias is challenging**
>
> We agree with the reviewer that detecting sycophancy bias presents significant challenges due to its reliance on understanding implicit and context-dependent human preferences. Despite these challenges, it is important to note that influence functions continue to perform relatively robustly compared to other baselines. For example, when matched at equal false positive rates, influence functions outperform GPT-4o, the strongest baseline, by a significant margin of 14.8% in sycophancy bias experiments. These results underscore the effectiveness of influence functions in managing more complex biases
>
> ---
>
> **[Q1] On more generalized validation samples**
>
> Thank you for your insightful comment. We acknowledge that requiring targeted evaluation sets can be a barrier to using our method (as mentioned in Remark 4.1 of the original draft). However, it is important to highlight that influence functions can still be effective even with less curated sets. To demonstrate this, we conducted additional experiments on bias detection using the original validation set of the Anthropic HH dataset, denoted as ‘Full’. This dataset was used in its unmodified, uncurated form as provided by the Anthropic HH validation set. The table below shows that influence functions effectively detect biased samples using this uncurated set compared to the high-quality, carefully filtered validation set used in our study (see Section 5.1.1 for more details).
>
> | | High-quality | Full |
> |:---:|:---:|:---:|
> |Length Bias|0.800|0.770|
> |Sycophancy Bias|0.711|0.585|
>
> These results confirm that influence functions are robust even without well-curated validation sets, demonstrating their utility with more generalized samples. We also note that influence functions demonstrate reasonable performance even with relatively small validation sets (approximately 50-100 samples), as shown in Figure 8 and Appendix G. This suggests the potential for practical applications, using small-scale expert data.
>
> We have revised Figure 4 (with Section 5.1.2) of our draft to reflect these findings, with the changes highlighted in red.
>
> ---
>
> **[Q2] Scalability of current method**
>
> **Scalability with increasing dataset sizes**
> As shown in Figure 5 of our original draft, time consumption scales linearly with dataset sizes up to 100k samples on a single NVIDIA A6000 GPU. This linear scaling is expected to extend to even larger datasets, as the primary computational cost lies in the backpropagation step, which is proportional to dataset size. Importantly, this assumes adequate storage for the compressed gradient vectors of the training and validation datasets. By employing OPORP, we have compressed gradient vectors to $2^{16}$ dimensions, facilitating the storage of millions of samples, e.g., 5M samples require approximately 1TB of storage. If even larger datasets are considered, smaller compression dimensions could be used to further reduce storage needs, though at the cost of increased estimation variance, which, according to the OPORP paper, is inversely proportional to the compressing dimension $\text{K}$ ($\text{Var} \propto 1/\text{K}$) [1].
>
> **Scalability with increasing model size**
> The computational expense also escalates with the model size due to the scaling of backpropagation costs with the number of model parameters. A trade-off here is the increased approximation error in DataInf, which arises from the need to approximate the Hessian matrix. This error potentially impacts performance of influence functions, particularly in scenarios demanding high precision.
>
>
> [1] Ping Li and Xiaoyun Li. Oporp: One permutation+ one random projection. In ACM SIGKDD Conference on Knowledge Discovery and Data Mining, 2023.

---

> ### Author Response · Authors · 2024-11-25
> **Gentle Reminder**
>
> Dear Reviewer roz9,
>
> Thank you again for your time and efforts in reviewing our paper.
>
> As the discussion period draws close, we kindly remind you that two days remain for further comments or questions. We would appreciate the opportunity to address any additional concerns you may have before the discussion phase ends.
>
> Thank you very much!
>
> Many thanks,
> Authors

---

> > ### Comment · Reviewer_roz9 · 2024-11-25
> > **Reply to Authors' responses**
> >
> > Thank you for the author’s response, which addressed my previous concerns. I have updated my score accordingly.

---

> ### Author Response · Authors · 2024-11-25
>
> Dear Reviewer roz9,
>
> Thank you for your response. We are pleased to hear that your concerns have been resolved and sincerely appreciate your decision to raise the score.
>
> If there are any remaining concerns or suggestions for improvement, we would be more than happy to address them.
>
> Best regards, Authors

---

### Author Response · Authors · 2024-11-21
**General Response to Reviewers**

**We greatly appreciate all reviewers for their time and effort in reviewing our paper and providing thoughtful suggestions.**

As most reviewers highlighted, our work addresses the practical and important topic of enhancing transparency and interpretability in human feedback by measuring their effect on reward modeling. We propose compute-efficient methods to enable applying influence functions to modern reward models with billions of parameters. Our experiments show that influence functions show convincing performance in detecting labeler bias existing in preference datasets. Below is a list of common strengths of our work that reviewers have kindly noted, and a summary of updates we have made in our revised draft.

---

## Common strengths
* Topic is practical and important (Reviewer Wn7U)
  * Enhances transparency (Reviewer CzsY)
  * Promising direction of interpretability (Reviewer roz9)
  * Offer insights into how human feedback shapes the reward model's outcomes (Reviewer 71Vm)
* Propose compute efficient methods (Reviewer roz9, CzsY, Wn7U, and facp)
* The paper is well written (Reviewer roz9, CzsY, and dPes)
* The experiments are well-designed (Reviewer CzsY and dPes)
  * Two potential applications (Reviewer dPes and facp)
* Performance is convincing (Reviewer CzsY, Wn7U, and dPes)

---

## Major updates in the revised draft

**The updates in our revised draft are color-coded in red for distinction.**

* We have updated Figure 4 to include influence function performance on an uncurated validation set (denoted as ‘full’), showing reasonable performance. Results show that influence functions can be used even in less-controlled conditions.

* We have updated Appendix E to report additional metrics for our bias detection experiment: area under the precision-recall curve (AP) and the TNR value at TPR=0.8 (TNR80), along with precision-recall curves. Influence functions outperform all baselines on these additional metrics.

* We have updated Appendix A.2 to include an experiment verifying whether the gradient compression step in our method affects performance, by comparing our estimates with DataInf.  Results show no performance degradation or significant difference in influence values.

* We have updated Appendix H.2 to include ablation experiments of few-shot prompting of LLM-based detectors. Results show that LLMs fail to successfully detect biased samples even when supplied with numerous examples of up to 50.

* We have revised the expression of our draft for clarity.

---

### Meta-Review · Area_Chair_MYyB · 2024-12-20

**Metareview:**

This paper proposes a novel approach to evaluate and refine reward models in RLHF by utilizing influence functions. The authors argue that human feedback is an indispensable component in training reward models for LLMs, but it may be noisy and biased, potentially leading to inconsistent rewards. To address this issue, they introduce a computationally efficient method to approximate influence functions, enabling the measurement of each data point’s impact on the reward model’s performance.

**Additional Comments On Reviewer Discussion:**

The reviewer's scores for this paper showed significant variance. To ensure an effective decision, the AC considered each reviewer’s comments and carefully read the manuscript. The outcome revealed substantial concerns regarding the paper, primarily about its novelty. Additionally, the dependency of influence functions on the dataset is considerable. Upon further examination of the current manuscript, I found a lack of ablation studies on the dataset and insufficient evidence of balanced performance across multiple maintenance scenarios.

In summary, based on the reviewers’ feedback, including their scores, confidence, and detailed comments, I believe the paper’s proposed method lacks sufficient novelty. I encourage the authors to incorporate the reviewers’ suggestions to improve the manuscript’s quality. Consequently, this paper is rejected.

---

### Decision · Program_Chairs · 2025-01-22

Reject